# A molecular switch regulating transcriptional repression and activation of PPARγ

Jinsai Shang [1], Sarah A. Mosure[1,2,3], Jie Zheng[1,4], Richard Brust[1], Jared Bass[1], Ashley Nichols[5], Laura A. Solt[3,6], Patrick R. Griffin [1,6] & Douglas J. Kojetin [1,6 ✉]

Nuclear receptor (NR) transcription factors use a conserved activation function-2 (AF-2) helix 12 mechanism for agonist-induced coactivator interaction and NR transcriptional activation. In contrast, ligand-induced corepressor-dependent NR repression appears to occur through structurally diverse mechanisms. We report two crystal structures of peroxisome proliferator-activated receptor gamma (PPARγ) in an inverse agonist/corepressor-bound transcriptionally repressive conformation. Helix 12 is displaced from the solvent-exposed active conformation and occupies the orthosteric ligand-binding pocket enabled by a conformational change that doubles the pocket volume. Paramagnetic relaxation enhancement (PRE) NMR and chemical crosslinking mass spectrometry confirm the repressive helix 12 conformation. PRE NMR also defines the mechanism of action of the corepressor-selective inverse agonist T0070907, and reveals that apo-helix 12 exchanges between transcriptionally active and repressive conformations—supporting a fundamental hypothesis in the NR field that helix 12 exchanges between transcriptionally active and repressive conformations.

[1] Department of Integrative Structural and Computational Biology, The Scripps Research Institute, Jupiter, FL 33458, USA. [2] Skaggs Graduate School of Chemical and Biological Sciences, The Scripps Research Institute, Jupiter, FL 33458, USA. [3] Department of Immunology and Microbiology, The Scripps Research Institute, Jupiter, FL 33458, USA. [4] Shanghai Institute of Materia Medica, Chinese Academy of Sciences, Shanghai 201203, China. [5] Summer Undergraduate Research Fellows (SURF) program, The Scripps Research Institute, Jupiter, FL 33458, USA. [6] Department of Molecular Medicine, The Scripps Research Institute, Jupiter, FL 33458, USA. ✉email: dkojetin@scripps.edu

Nuclear receptor (NR) transcription factors recruit chromatin remodeling transcriptional coactivator and corepressor proteins in a ligand-dependent manner. NRs are modular domain proteins containing an N-terminal disordered activation function-1 (AF-1) region, a central DNA-binding domain, and C-terminal ligand-binding domain (LBD) containing the ligand-dependent activation function-2 (AF-2) coregulator binding surface. Structural studies of many different NRs have revealed a unified mechanism describing how agonist ligands activate NR transcription[1,2]. Agonist binding to the NR orthosteric ligand-binding pocket stabilizes the C-terminal structural element in the LBD, helix 12, in an active conformation that facilitates binding of LXXLL-containing motifs present in transcriptional coactivator proteins. In contrast, structural studies available for only a few NRs have not revealed a unified mechanism for repression of NR transcription[3–14], suggesting that corepressor-dependent transcriptional repression of NRs occurs through structurally diverse mechanisms. Furthermore, the conformational ensemble of helix 12 in apo-NR LBDs, which is hypothesized to exchange between a transcriptionally active conformation and a repressive conformation, remains poorly defined. Although crystal structures of apo-NR LBDs have revealed active and non-active helix 12 conformations[15–19], these structures—some of which are stabilized by crystal contacts between symmetry-related molecules within the crystal lattice, or by mutations or coregulator peptides that reduce helix 12 dynamics to facilitate crystal formation for structure determination—are not fully representative of the dynamic NR LBD conformational ensemble.

Proliferator-activated receptor gamma (PPARγ), a NR that regulates gene programs that influence cellular metabolism, differentiation, and insulin sensitization[20], is regulated by endogenous cellular lipids and synthetic ligands, including the thiazolidinedione (TZD) class of antidiabetic drugs. X-ray crystallography, NMR spectroscopy, and hydrogen-deuterium exchange (HDX) mass spectrometry studies have revealed how PPARγ agonists and coactivator peptides influence the transcriptionally active conformation of PPARγ[16,21–26]. However, a structural understanding of the repressive conformation of PPARγ capable of recruiting corepressor proteins that influence PPARγ transcription[27] has remained elusive in part due to the lack of a ligand capable of stabilizing a corepressor-bound state.

We recently reported that a synthetic covalent PPARγ ligand called T0070907[28] used in the field as a transcriptionally neutral PPARγ antagonist actually functions as a transcriptionally repressive corepressor-selective PPARγ inverse agonist[29]. Here we report two crystal structures of PPARγ bound to T0070907 and corepressor peptides that reveal a unique a transcriptionally repressive conformation compared with all other reported structures of corepressor-bound NRs. A comparative structural analysis to the transcriptionally active conformation and the apo-PPARγ conformational ensemble using solution paramagnetic relaxation enhancement (PRE) NMR validates the unique repressive conformation and provides evidence that helix 12 in an apo-NR exchanges between transcriptionally active and repressive conformations.

## Results

### Corepressor ID2 motif is sufficient for PPARγ interaction. The NCoR and SMRT corepressor proteins contain two conserved interaction domain (ID) regions, ID1 and ID2. A fluorescence polarization coregulator peptide profiling assay reveals the corepressor ID2 motif robustly interacts with apo-PPARγ LBD (Fig. 1a). In a mammalian two-hybrid protein–protein interaction cellular assay (Fig. 1b), the interaction between the PPARγ LBD

and the NCoR corepressor receptor interaction domain (RID) is dependent on the ID2 corepressor motif and enhanced by the corepressor-selective inverse agonist T0070907. In a cellular reporter assay, T0070907 represses transcription of full-length PPARγ (Fig. 1c) and represses the expression of PPARγ target genes in 3T3-L1 preadipocytes[29]. Consistent with the mammalian two-hybrid showing the ID2 sequence is necessary for PPARγ interaction, in a TR-FRET biochemical peptide interaction assay (Fig. 1d) T0070907 enhances the binding of corepressor ID2 peptides to the PPARγ LBD and decreases binding of coactivator peptide. These profiles are opposite to the TZD PPARγ agonist rosiglitazone, which increases PPARγ transcription, decreases interaction of corepressor ID2 peptides, and increases interaction of TRAP220 coactivator ID2 motif peptide.

Using NMR spectroscopy, we previously showed that T0070907-bound PPARγ LBD slowly exchanges between two long-lived conformations, including a conformation similar to the corepressor-bound state that binds NCoR ID2 peptide with high affinity[29] and stabilizes a unique, single helix 12 conformation in solution[30]. We therefore posited that T0070907 could be used to facilitate crystallization of the PPARγ LBD bound to peptides derived from NCoR and SMRT ID2 motifs.

### Crystal structures of the repressive conformation of PPARγ. We obtained crystals for T0070907-bound PPARγ LBD in the presence of NCoR and SMRT ID2 peptides under the same crystallization condition, but each complex crystallized in different space groups and with a different number of molecules in the asymmetric unit, one and two, respectively. We solved the crystal structures of T0070907-bound PPARγ LBD with NCoR ID2 (Fig. 2a, Supplementary Fig. 1) and SMRT ID2 (Fig. 2b, Supplementary Fig. 2) peptides at 1.8 and 2.1 Å, respectively (Supplementary Table 1). We also solved the crystal structure of rosiglitazone-bound PPARγ LBD with TRAP220 coactivator peptide to 2.3 Å resolution (Fig. 2c, Supplementary Fig. 3, Supplementary Table 1) to compare the transcriptionally active conformation with the corepressor-bound transcriptionally repressive conformations.

The transcriptionally active PPARγ conformation bound to the agonist rosiglitazone and TRAP220 coactivator peptide (Fig. 2c), which is similar to the active conformation bound to SRC-1 peptide[16], is defined by the association of three helical elements—helix 3, helix 4/5, and the critical regulatory element helix 12—that form the docking surface for the coactivator peptide. Rosiglitazone binds in the orthosteric ligand-binding pocket (Fig. 3a) and forms hydrogen bonds with residues near the AF-2 surface including the side chain phenolic hydroxyl group of Y473 on helix 12, which stabilizes helix 12 into an active conformation. Helix 3 forms an extended linear conformation connecting the flexible Ω-loop to helix 2b and the β-sheet via β-strand 1.

In the transcriptionally repressive PPARγ conformation bound to the corepressor-selective inverse agonist T0070907 and NCoR or SMRT ID2 peptides (Fig. 2a, b, respectively), there are several notable structural changes compared with the active conformation (Fig. 3c). The corepressor peptide binds to the AF-2 surface through interactions with helix 3 and helix 4/5, but unlike the active conformation the peptide does not contact helix 12. As observed in other crystal structures of corepressor-peptide-bound NR LBDs[3–9,11–14], there is an extension of the corepressor peptide by one helical turn relative to the coactivator peptide that would physically clash with helix 12 in the active conformation. Instead, helix 12 occupies the orthosteric ligand-binding pocket overlapping with the active conformation rosiglitazone orthosteric binding mode (Fig. 3b), flanked one side by helix 3 and on the other by helix 2b and the β-sheet. This repressive helix 12

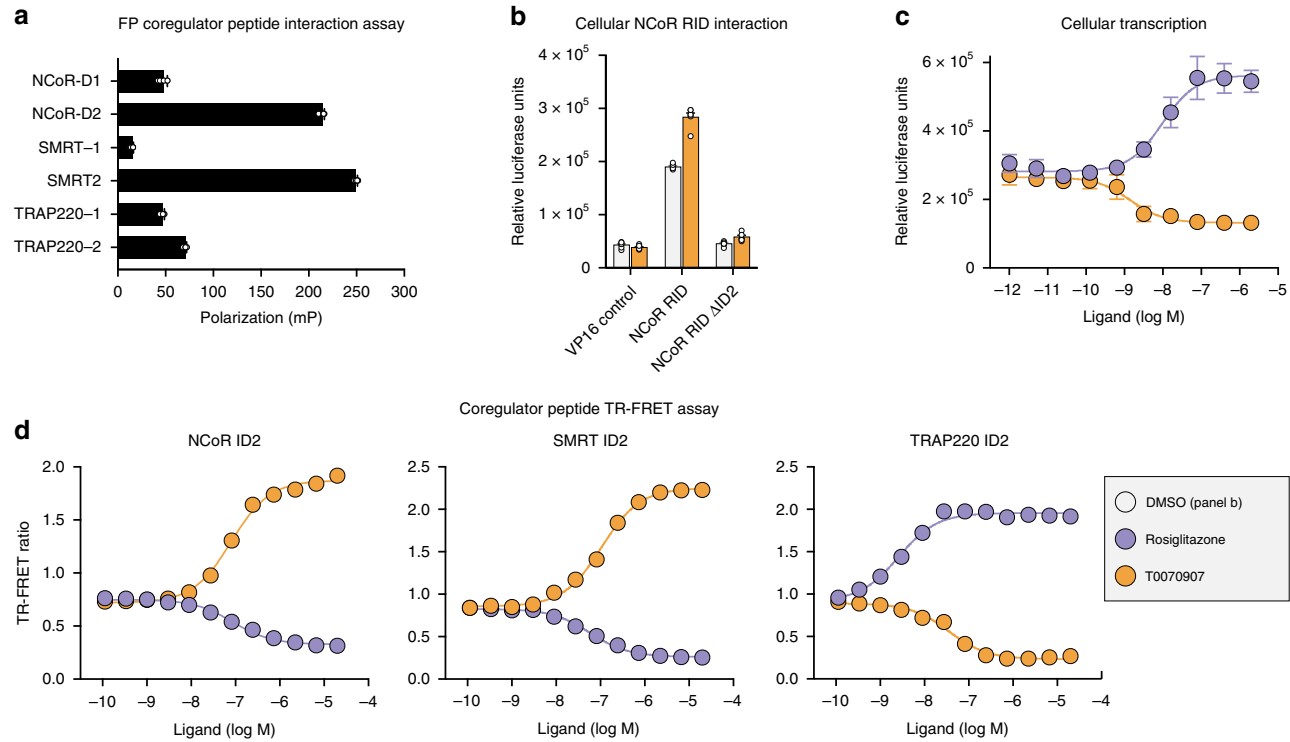

**Fig. 1 PPARγ interacts with the transcriptional corepressor ID2 motif. a** Fluorescence polarization coregulator interaction assay shows a robust interaction between PPARγ LBD with corepressor ID2 peptides from NCoR and SMRT as well as a TRAP220 coactivator peptide ($n = 6$). **b** Mammalian two-hybrid assay in HEK293T cells measuring the effect of T0070907 (10 μM) on the interaction between the Gal4-PPARγ LBD and the VP16-NCoR RID ($n = 6$; mean ± s.d.). **c** Luciferase transcriptional reporter assay measuring the ligand-dependent change in PPARγ transcription ($n = 4$; mean ± s.d.). **d** TR-FRET biochemical assay measuring the ligand-dependent change in the interaction of PPARγ LBD with SMRT ID2, NCoR ID2, and TRAP220 ID2 peptides ($n = 3$; mean ± s.e.m.). Source data are provided as a Source Data file.

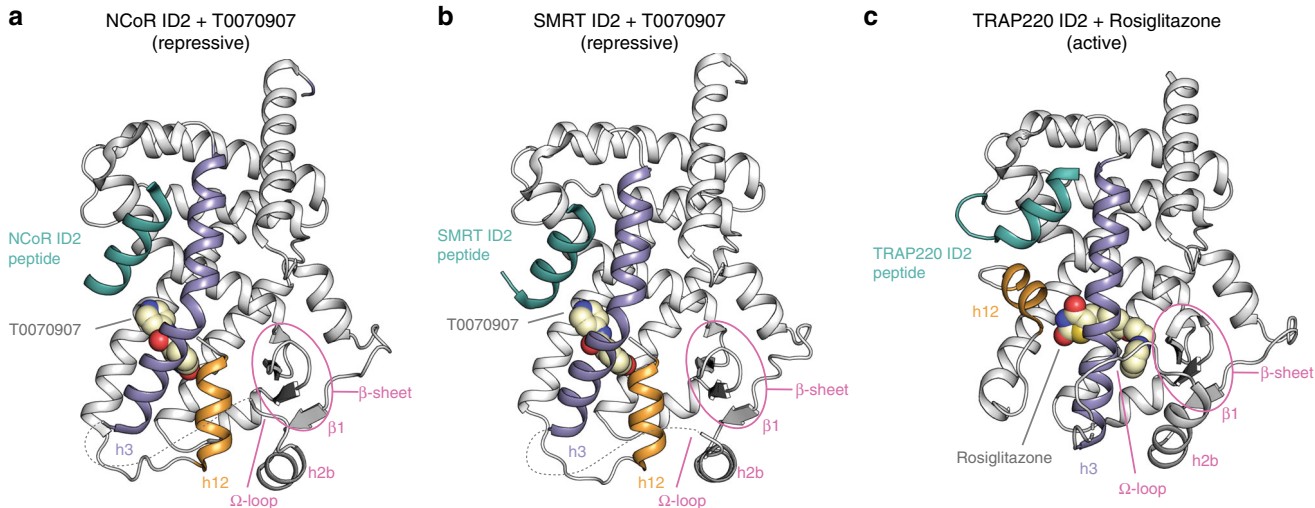

**Fig. 2 Structure of PPARγ LBD bound to T0070907 and corepressor peptides. a** Crystal structure of PPARγ LBD bound to the corepressor-selective inverse agonist T0070907 and NCoR ID2 peptide (PDB 6ONI). **b** Crystal structure of PPARγ LBD bound to the corepressor-selective inverse agonist T0070907 and SMRT ID2 peptide (PDB 6PDZ). **c** Crystal structure of PPARγ LBD bound to the coactivator-selective agonist rosiglitazone and TRAP220 ID2 peptide (PDB 6ONJ).

conformation occupies part of the space occupied by the N-terminal region of helix 3 in the active conformation, facilitated by a kink in helix 3 resulting in a ~40° pivot of the N-terminus of helix 3 (residues E276–R288) and a ~5 Å displacement away from helix 2b and the β-sheet toward helix 11. Furthermore, the insertion of helix 12 into the pocket results in a shift in helix 2b and β-strand 1 away from the pocket relative to the active

conformation. The conformational change in helix 3, which moves it toward the active helix 11 conformation, is also associated with a conformational change in helix 11 (25° pivot, 3 Å displacement) away from the pocket relative to the active helix 11 conformation. In total, these conformational changes expand the repressive conformation ligand-binding pocket to a volume of ~3100 Å³, double the ~1600 Å³ volume of the active

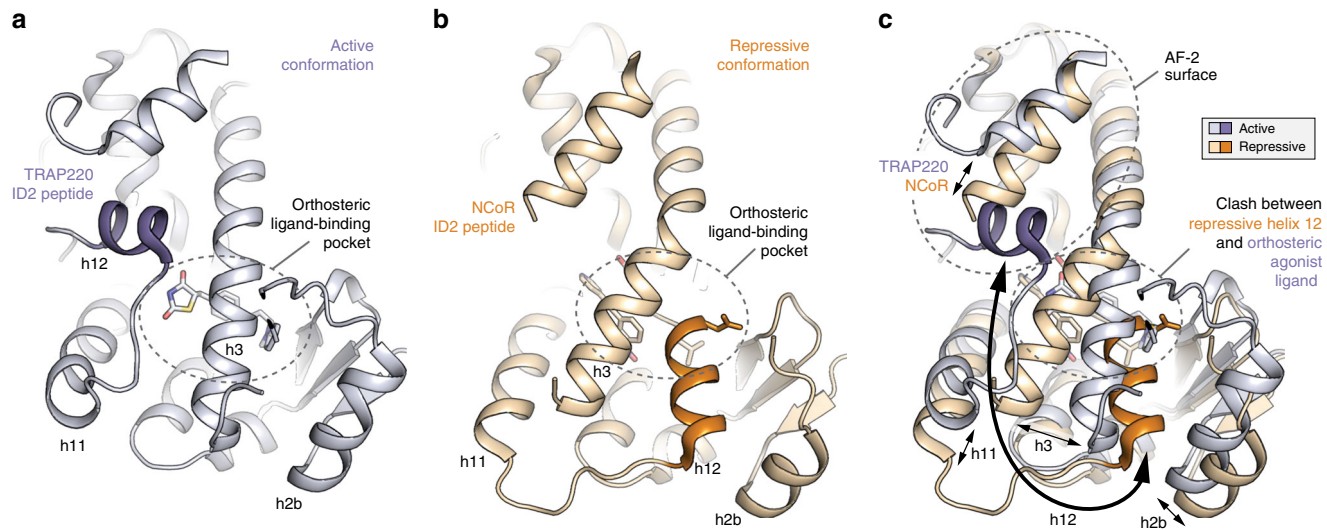

**Fig. 3 Structural changes between the repressive and active PPARγ LBD conformations. a**, **b** Helical structural elements that show notable conformational changes in the crystal structures of **a** PPARγ LBD bound to rosiglitazone and TRAP220 ID2 peptide (PDB 6ONJ) in the active conformation and **b** PPARγ LBD bound to T0070907 and NCoR ID2 peptide (PDB 6ONI) in the repressive conformation. **c** Structural overlay of the active and repressive conformation PPARγ LBD crystal structures with arrows depicting the movement of the helical structural elements.

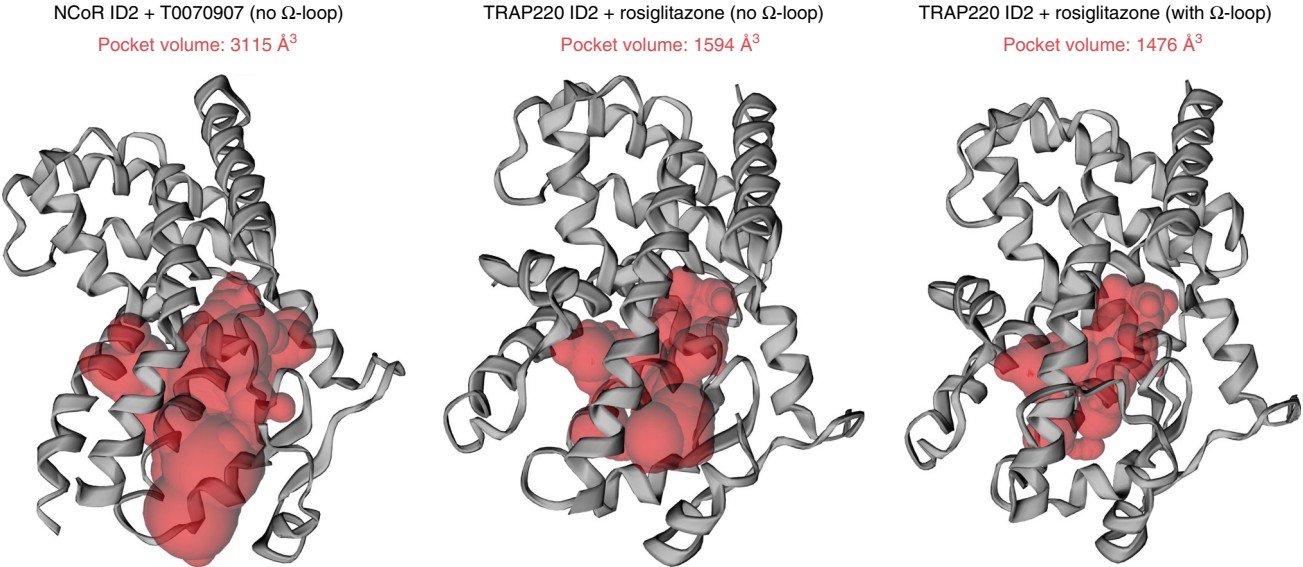

**Fig. 4 Ligand-binding pocket volume doubles in the repressive conformation.** Ligand-binding pocket volumes (red surfaces) were calculated and displayed using the program CASTp. Ligands were removed from the structures for all calculations. Helix 12 was removed from the repressive conformation structure (PDB 6ONI) to assess the relative pocket volume to the active conformation structures. Calculations for the active conformation structure (PDB 6ONJ), which displays density for the Ω-loop region, were performed with and without the Ω-loop region for comparison to the repressive conformation structure, which lacks the Ω-loop region.

conformation ligand-binding pocket (Fig. 4), enabling helix 12 in the repressive conformation to occupy the orthosteric ligand-binding pocket.

**Structural features of corepressor interaction interface.** Direct and water-mediated interactions involving six residues in the PPARγ AF-2 surface facilitate binding of the corepressor peptide (Fig. 5a). This includes one of the two "charge clamp" residues, K301 important for binding coactivator peptides, as well as Q286 and Q294 in helix 3, Q314 in helix 4, and N312 and K319 in helix 5; the other charge clamp residue, E471 on helix 12, is displaced into the pocket in the repressive conformation. In the active conformation, fewer interactions facilitate binding of the

coactivator peptide (Fig. 5b). The charge clamp residues, K301 and E471 in helix 3 and helix 12, respectively, mediate the primary interactions with the coactivator peptide helix. However, the extended N-terminal loop of the coactivator peptide interacts with N312 in helix 5, G399 in the helix 8–9 loop, and K474 on helix 12. This extended N-terminal coactivator interaction with the helix 8–9 loop to our knowledge has not been described in other published coactivator peptide-bound PPARγ crystals structures.

To determine the functional role of the corepressor-peptide-interacting residues, we mutated residues with side chain contacts to the NCoR ID2 peptide to alanine and tested the effect of these structure-guided mutants using a fluorescence polarization peptide interaction assay with apo- and T0070907-bound PPARγ

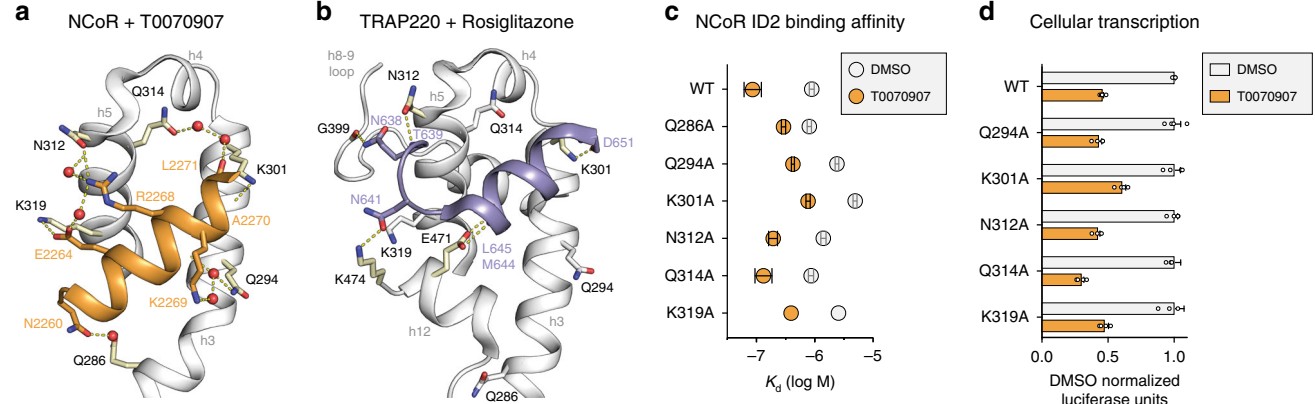

**Fig. 5 Comparison of the corepressor and coactivator interaction network. a** Interactions with six residues in the PPARγ AF-2 surface mediate binding to the helical core of the NCoR ID2 peptide (PDB 6ONI). **b** Fewer interactions mediate binding to the helical core of the TRAP220 coactivator peptide (PDB 6ONJ). **c** NCoR ID2 peptide affinities and errors (mean ± s.d. of the fit to a one site—total binding equation) from fluorescence polarization peptide interaction assays using structure-guided mutants (Supplementary Fig. 4) displayed to illustrate the difference in NCoR ID2 peptide binding affinity between apo- and T0070907-bound PPARγ LBD. **d** Cell-based luciferase reporter assay in HEK293T cells treated with DMSO control or 5 µM T0070907 measuring the ligand-dependent change in wild-type or mutant PPARγ transcription; data normalized to each WT or mutant construct vehicle control (DMSO) condition ($n = 4$; mean ± s.e.m.). Source data are provided as a Source Data file.

LBD (Supplementary Fig. 4). We also tested structure-guided alanine mutants of the coactivator peptide-interacting residues along with a K301A/E471A charge clamp double mutant—both residues are involved in coactivator interaction but only K301 is involved in corepressor interaction—on the interaction between the TRAP220 coactivator ID2 peptide and apo-PPARγ LBD (Supplementary Fig. 5). The K301A mutant displays the largest negative effect on corepressor-peptide binding affinity in the apo- and T0070907-bound forms, though the difference was <1 order of magnitude (Fig. 5c). In contrast, the charge clamp mutants shows a more pronounced negative impact on the coactivator peptide binding affinity, up to 2 orders of magnitude (Supplementary Fig. 5). All of the corepressor structure-guided mutants retained corepressor-selective responsiveness to T0070907; the affinity change for the NCoR ID2 peptide is similarly increased in the T0070907-bound mutants (Fig. 5c) indicating that T0070907 enhances corepressor affinity without significantly affecting these corepressor-interacting residues. These corepressor interaction interface structure-guided mutants support the PPARγ crystal structures whereby many interactions contribute to binding corepressor peptide, whereas the coactivator interaction is predominately mediated by the "charge clamp" residues.

**Repressive T0070907 binding mode and helix 12 conformation**. There are several interesting features of the T0070907 ligand-binding pose in the repressive conformation. An extensive network of interactions between T0070907 and nearby residues appear to lock helix 12 within the orthosteric ligand-binding pocket (Fig. 6a). These include several polar π-stacking interactions occur between the T0070907 pyridyl group and nearby aromatic side chains including H323 on helix 5, H449 on helix 11, and the C-terminal residue Y477 that extends from helix 12 inside the orthosteric ligand-binding pocket. Furthermore, a network of hydrogen bond and electrostatic interactions also contributes to the repressive T0070907 binding pose, including M364 and K367 on helix 3 to the nitro group of T0070907 and the C-terminal carboxylate of Y477, which also interacts with the amide in T0070907 and the hydroxyl group of Y327. Focusing on helix 12 within the orthosteric pocket (Fig. 6b), a network of direct and water-mediated hydrogen bonds and electrostatic interactions connect helix 12 (Q470, E471, and K474) to helix 3 (R280 and R288), and S342 on the β-sheet to helix 12 via D475.

We generated mutants of these residues and tested them using a TR-FRET assay that measures the ligand-dependent effect on corepressor-peptide interaction (Fig. 6c–f). Focusing on residues that directly interact with T0070907 (Fig. 6c), two mutants that are part of the nitro group interaction network, Y327A and K367A, show no change in corepressor-peptide interaction. Ligand displacement and mass spectrometry data reveal that T0070907 does not covalently bind to these mutants (Supplementary Fig. 6 and Supplementary Table 2), implicating Y327 and K367 in the covalent attachment mechanism. Other T0070907-interacting mutants show decreased T0070907 $EC_{50}$ values and/or TR-FRET windows, indicating the mutants affect binding efficiency of T0070907 or ligand-induced change in corepressor-peptide affinity, respectively. The C-terminal Y477A mutant and truncated mutants lacking Y477 (L476*) or helix 12 and the helix 11–12 loop (T461*) all show either no response to T0070907 or a slight decrease in TR-FRET window with similar $EC_{50}$ values to wild-type (WT) receptor (Fig. 6d), indicating T0070907 still covalently binds but the mutations and truncations perturb the repressive T0070907 binding mode and/or repressive helix 12 conformation within the orthosteric pocket. We also tested a three residue C-terminal extension (+GAP) mutant, which we posited may protrude from the AF-2 surface and impact corepressor-peptide interaction and remove the combined Y477 π-stacking/C-terminal carboxylate interaction network. This +GAP mutant also shows relatively no change in TR-FRET window while retaining the ability to covalently bind T0070907 (Supplementary Fig. 6). Most of the mutations designed to disrupt the protein interaction network in the repressive helix 12 conformation including residues on helix 3 (Fig. 6e) and helix 12 (Fig. 6f) decrease the T0070907-dependent increase in corepressor-peptide interaction without affecting covalent binding of T0070907 (Supplementary Fig. 6).

To complement the TR-FRET coregulator peptide interaction assays, we performed fluorescence polarization peptide interaction assays to determine how the mutations affect the affinity for binding the NCoR ID2 peptide. Overall, the FP-derived affinity trends are consistent with the TR-FRET data (Fig. 6g and Supplementary Fig. 7). For example, the Y327A mutant, K367 mutant, Y477A mutant, L476* truncation, and +GAP extension all show no increase in NCoR ID2 affinity in the presence of T0070907, and many of the others show diminished

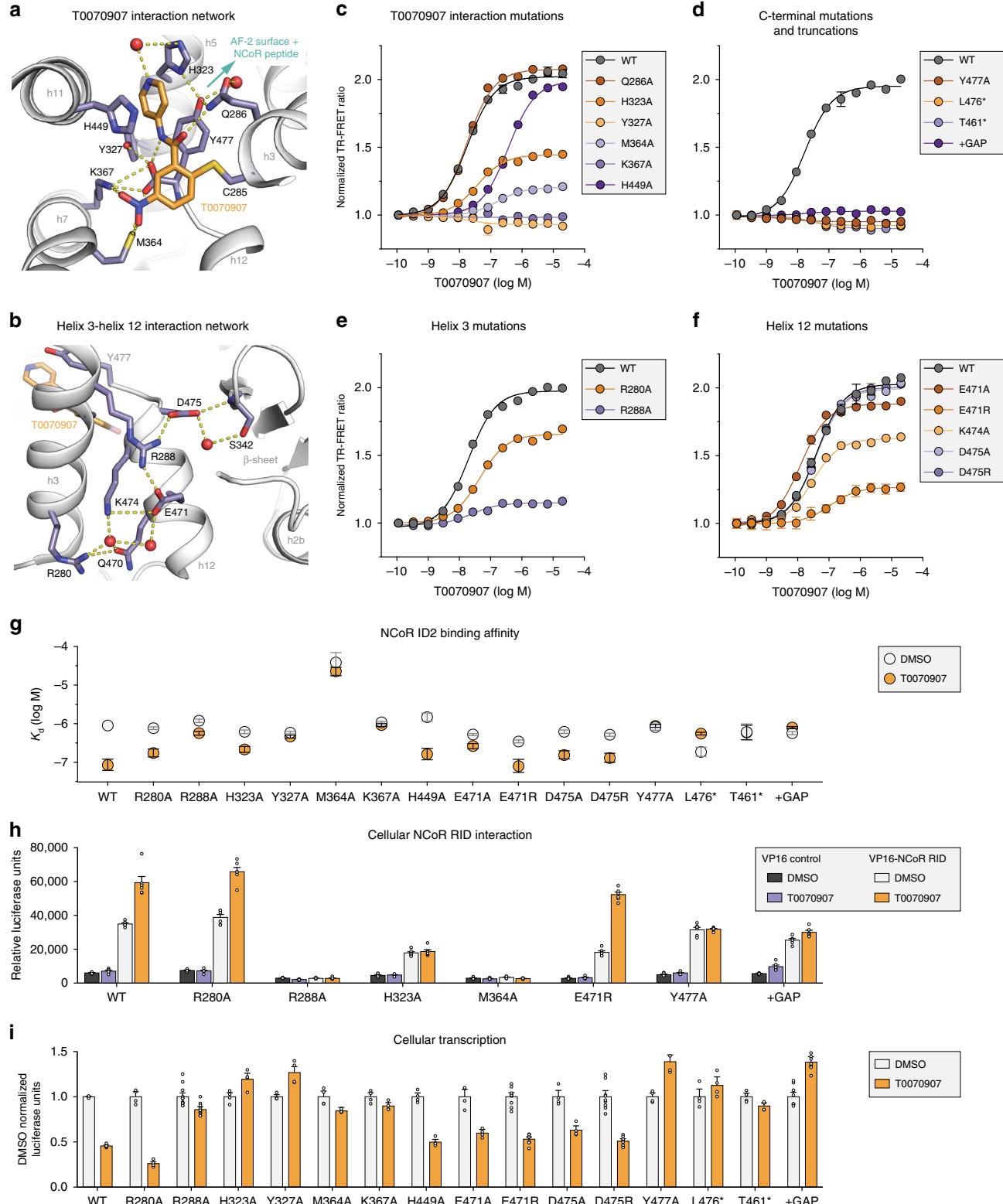

corepressor-selective responsiveness to T0070907. Interestingly, the M364A, which is buried in the pocket, mutant significantly weakens NCoR ID2 peptide affinity in both the apo- and T0070907-bound forms. The M364 thiol interacts with both the T0070907 nitro group (3.3 and 4.3 Å) and the C-terminal carboxylate of Y477 (3.5 Å), the latter of which may be important for stabilizing the repressive helix 12 conformation in the orthosteric pocket.

We confirmed the TR-FRET and FP biochemical peptide-based findings for several of the mutants using a mammalian two-hybrid protein–protein interaction cellular assay to assess the interaction between the PPARγ LBD and the entire NCoR RID (Fig. 6h), revealing similar trends to the biochemical data. Two of the mutants (R288A and M364A) show no basal or T0070907-dependent interaction with NCoR RID. Furthermore, the Y477A mutant and +GAP extension maintain basal NCoR RID

**Fig. 6 Mutagenesis validates the inverse agonist-dependent repressive helix 12 conformation. a** Interactions between PPARγ and the corepressor-selective inverse agonist T0070907 (PDB 6ONI). **b** The repressive helix 12 conformation is mediated through a network of interactions with residues on helix 3 (PDB 6ONI). **c, d** Structure-guided mutants of the T0070907 interaction tested using a TR-FRET biochemical assay measuring the ligand-dependent change in the interaction of PPARγ LBD with a peptide derived from the NCoR corepressor ($n = 3$; mean ± s.e.m.). **e, f** Structure-guided mutants of the helix 3/helix 12 interaction network tested using the TR-FRET biochemical assay ($n = 3$; mean ± s.e.m.). **g** NCoR ID2 peptide affinities and errors (mean ± s.d. of the fit to a one site—total binding equation) from fluorescence polarization peptide interaction assays using structure-guided mutants (Supplementary Fig. 4) displayed to illustrate the difference in NCoR ID2 peptide binding affinity between apo- and T0070907-bound PPARγ LBD. **h** Mammalian two-hybrid assay measuring the effect of T0070907 (10 μM) on the interaction between the Gal4-PPARγ LBD and the VP16-NCoR RID ($n = 6$; mean ± s.e.m.). **i** Luciferase transcriptional reporter assay in HEK293T cells treated with DMSO control or 5 μM T0070907 measuring the ligand-dependent change in wild-type or mutant PPARγ transcription; data normalized to each WT or mutant construct vehicle control (DMSO) condition ($n = 4$–12; mean ± s.e.m.). Source data are provided as a Source Data file.

interaction but lost T0070907-dependent increase in the interaction. We also performed a cellular reporter assay to assess the impact of the mutations on the transcription of full-length PPARγ (Fig. 6i). There is generally good agreement between the biochemical and cellular NCoR interaction profiles with the ability of T0070907 to repress the transcription of the mutant variants. For example, the mutants that show relatively no T0070907-dependent change in NCoR ID2 peptide binding affinity either do not show T0070907-dependent transcriptional repression or in many cases show activation in the presence of T0070907, including H323A, Y327A, M364A, K367A, Y477A, L476* truncation, T461* truncation, and the +GAP extension. In contrast, mutants that show a T0070907-dependent increase in NCoR ID2 peptide binding affinity generally show T0070907-dependent transcriptional repression.

Taken together, data from these structure-guided mutations provide confirmation for the repressive T0070907 binding pose and repressive helix 12 conformation within the orthosteric pocket in the corepressor-peptide-bound PPARγ LBD crystal structures. Furthermore, in contrast to the corepressor interaction interface structure-guided mutants (Fig. 5), many of the T0070907 interaction network mutants (Fig. 6) reduce or ablate corepressor-selective responsiveness to T0070907. This suggests that the repressive helix 12 conformation within the orthosteric pocket may be the primary contribution to the T0070907-dependent increase in corepressor binding affinity and repression of PPARγ transcription.

**PRE NMR of the helix 12 conformational ensemble.** The repressive helix 12 conformation is unique among the several non-active PPARγ helix 12 conformations captured by crystallography[16,31,32], which are frequently referred to in the field as "inactive" although these conformations are likely an artifact influenced by crystal contacts (Supplementary Fig. 8). Although there are no apparent crystal contacts that would influence helix 12 conformation in our corepressor-peptide-bound structures—helix 12 is within the orthosteric ligand-binding pocket and not solvent exposed (Supplementary Fig. 9)—we used PRE NMR to validate that the helix 12 conformation captured in the repressive conformation crystal structure occurs in solution and is not a result of crystallization artifacts.

For PRE NMR of the repressive conformation, we used a mutant variant of the PPARγ LBD with a cysteine introduced on helix 12 (K474C) keeping the native cysteine intact (C285) for covalent attachment of T0070907. As we validated in a previous [19]F NMR study[30], when added in slight molar excess T0070907 covalently attaches to C285 but not K474C in this mutant variant. The subsequent addition of the nitroxide spin label MTSL allows site-specific covalent attachment to K474C, providing a PRE-sensitive spin label reporter of helix 12 conformation. We collected 2D [$^1$H,$^{15}$N]-TROSY-HSQC NMR data of

MTSL-labeled T0070907-bound [K474C]-PPARγ LBD in the presence of NCoR ID2 peptide in the paramagnetic state (oxidized MTSL; $I_{para}$) and diamagnetic state (reduced MTSL; $I_{dia}$) (Fig. 7a, b). In the PRE NMR data, NMR peaks corresponding to residues within close proximity of the MTSL spin label will show a decrease in PRE peak intensity ratio ($I_{para}/I_{dia}$) ranging from a PRE ratio of ~0 (<~13 Å distance to the MTSL label) or a ratio between 0 and 1 (~13–25 Å distance to the MTSL label)[33]. Indeed, there are notable PRE NMR effects for residues with well resolved NMR peaks on helix 3, the β-sheet, and helix 2b (Fig. 7c, d), indicating the crystallized repressive conformation of helix 12 is consistent with the solution-state conformation.

We also performed PRE NMR of the active helix 12 conformation of the PPARγ LBD bound to rosiglitazone and coactivator peptide, and the apo-helix 12 conformation (Fig. 7a, b). For these studies, we used a K474C/C285S double mutant protein that we also used previously for [19]F NMR[30], which enables attachment of MTSL to K474C on helix 12 only. PRE analysis in the active conformation using MTSL-labeled rosiglitazone-bound [K474C/C285S]-PPARγ LBD in the presence of TRAP220 ID2 peptide (Fig. 7c, d) reveals localized effects on helix 11, helix 12, and AF-2 surface residues on helix 5 (Y320, G321), consistent with the active conformation crystal structure. PRE analysis of the apo-conformation using MTSL-labeled [K474C/C285S]-PPARγ LBD (Fig. 7c, d) reveals predominate effects in the β-sheet and helix 2a consistent with a repressive-like helix 12 conformation with additional effects for residues in the AF-2 surface consistent with an active-like helix 12 conformation. The repressive-like helix 12-dependent PRE effects for apo-PPARγ occur "deeper" in the ligand-binding pocket compared with PPARγ bound to T0070907 and corepressor peptide (Fig. 7d). Notably, NMR analysis of apo-PPARγ residues in the AF-2 surface and ligand-binding pocket is limited because of intermediate exchange on the NMR time scale[25,26], which results in very broad or absent NMR peaks. To ensure there are no MTSL spin label-specific PRE effects to this analysis, we performed PRE NMR using a 3-Maleimdo-PROXYL spin label, which shows qualitatively similar PRE effects (Supplementary Fig. 10) to the MTSL data (Fig. 7). Furthermore, NMR overlays with non-spin labeled samples indicate that covalent attachment of the MTSL label causes mostly localized chemical shift perturbations (Supplementary Fig. 11). Taken together, these data not only validate the repressive and active conformation PPARγ LBD crystal structures (Fig. 2), but also indicate the apo-helix 12 exchanges between conformations similar to transcriptionally repressive (T0070907 and NCoR ID2 peptide bound) and active (rosiglitazone and TRAP220 ID2 peptide bound) states.

**Validation with chemical crosslinking mass spectrometry.** We recently reported a chemical crosslinking mass spectrometry (XL-MS) study on PPARγ LBD in the absence or presence of several ligands[34]—rosiglitazone (full agonist), MRL24 (partial agonist),

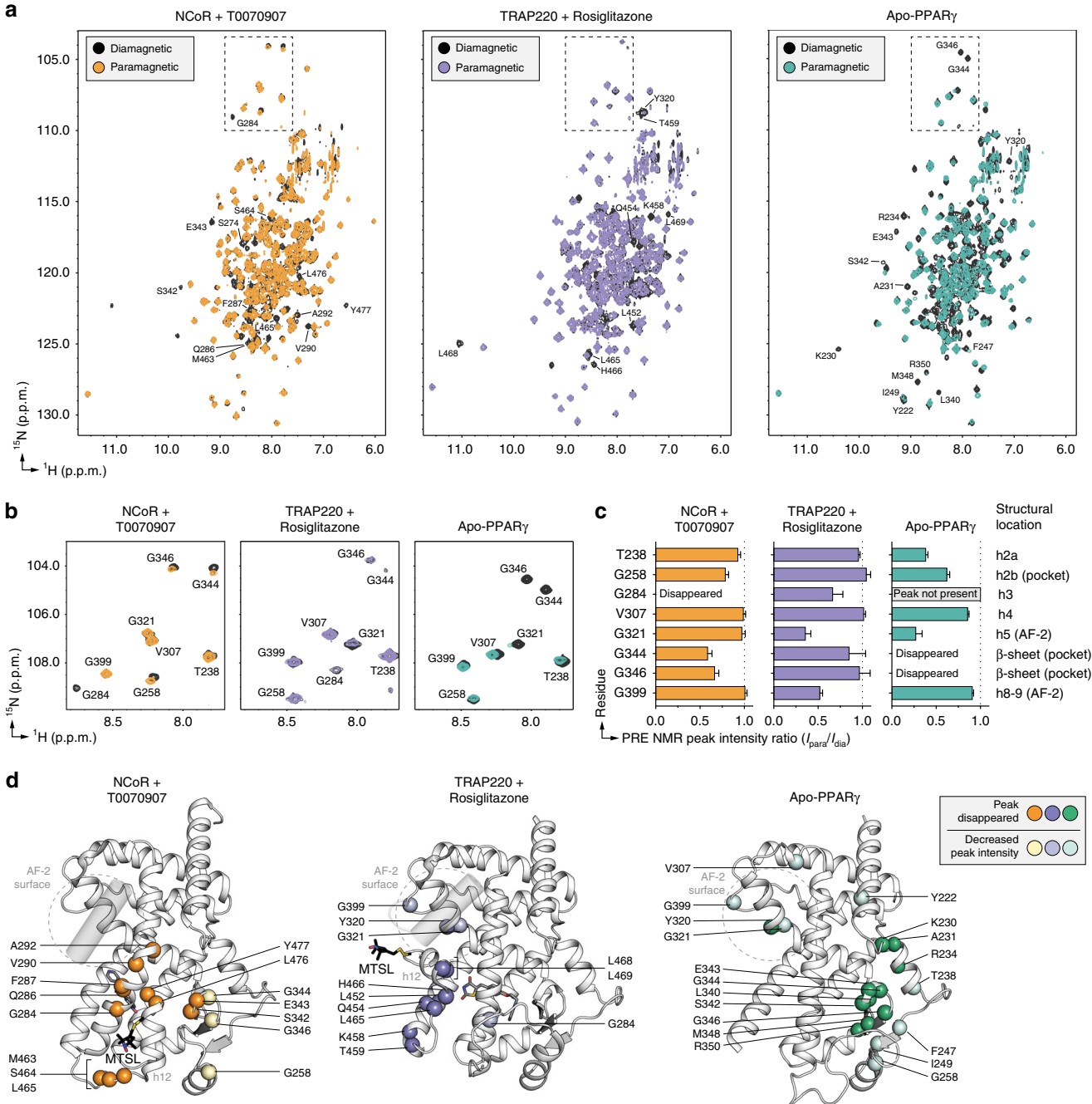

**Fig. 7 PRE NMR of the repressive, active, and apo-helix 12 conformations. a** Full 2D [$^1$H,$^{15}$N]-TROSY-HSQC spectra from the PRE analysis (see manuscript text and methods for sample descriptions). **b**, **c** PRE NMR data from (**b**) a well-dispersed upfield $^{15}$N region was used for (**c**) quantitative peak intensity analysis (error bars calculated using s.d. of data noise levels). **d** Structural mapping of PRE NMR data to the repressive (NCoR + T0070907; PDB 6ONI), active (TRAP220 + rosiglitazone; PDB 6ONJ), and apo (PDB 1PRG, chain A; helix 12, residues 460–477, is not shown as this conformation is not consistent with the PRE NMR data) conformation PPARγ LBD crystal structures. Source data are provided as a Source Data file.

SR11023 (antagonist), and GW9662 (covalent antagonist). An NCoR ID2 peptide-dependent increase in BS3 crosslink enrichment was detected between K474 on helix 12 with two residues in the Ω-loop (K265 and K275). No NCoR ID2 peptide-dependent BS3 crosslinks were detected between K474 and a residue located in the AF-2 surface, K301 on helix 3, which were detected for apo-PPARγ and SRC-1 ID2 peptide-bound forms. We previously used this XL-MS data to model the agonist and antagonist conformational landscape of helix 12, which included two modeled corepressor-bound conformations where helix 12 is surface exposed and partially wraps around helix 3 based on the K474-

K265 (Supplementary Fig. 12a) and K474-K275 (Supplementary Fig. 12b) crosslinks. However, at that time it was not apparent that helix 12 could adopt a corepressor-bound conformation within the orthosteric pocket. The K474-K265 XL-MS modeled helix 12 conformation is similar to two recent crystal structures of PPARγ LBD bound to two chemically related antagonist and inverse agonist ligands, SR11023 and SR10171, respectively, without coregulator peptide[31]. These structures also show surface exposed helix 12 conformations that wrap around helix 3 but in a different orientation (Supplementary Fig. 12c, d), a conformation that may be influenced by crystal contacts (Supplementary Fig. 8c).

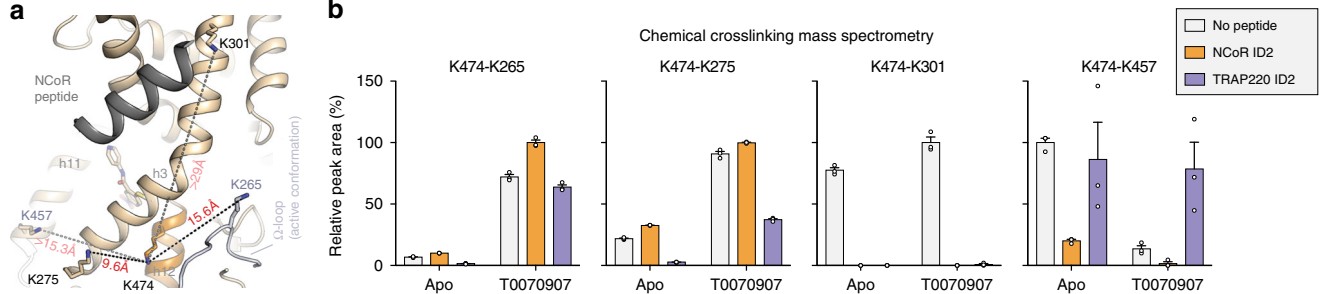

**Fig. 8 XL-MS confirms the repressive helix 12 conformation within the orthosteric ligand-binding pocket. a** Distances between K474 on helix 12 and several lysine residues structurally proximal to helix 12 (K265 and K275) or further away (K301 and K457) in the crystal structure of PPARγ LBD bound to T0070907 and NCoR ID2 peptide (PDB 6ONI). **b** Relative peak area of four K474-crosslinked peptides normalized to a control uncrosslinked peptide and to the highest mean peak area within each condition (mean ± s.e.m.; $n = 3$). Source data are provided as a Source Data file.

In our crystal structures of T0070907-bound PPARγ LBD bound to corepressor peptides, K474 on helix 12 is positioned such that the BS3 crosslinker (linker arm of 11.3 Å) would be predicted to crosslink the side chain of K474 on helix 12 to the side chains of K265 and K275 on the Ω-loop (Fig. 8a) with lysine side chain Nζ distances of 15.6 and 9.6 Å, respectively, given a ~3–6 Å tolerance for side chain flexibility[35]. However, the K474-K265 and K474-K275 distances in the SR11023- and SR10171-bound crystal structures range from 22 to 28 Å, which is ~11–18 Å longer than the BS3 linker arm distance (Supplementary Fig. 12c, d) and therefore not as compatible with the XL-MS data[34]. Furthermore, the XL-MS modeled conformations we generated[34] are compatible with an individual K474-K265 or K474-K275 crosslink (Supplementary Fig. 12a, b), but unlike our repressive conformation crystal structure these modeled conformations do not satisfy both crosslinks simultaneously.

To directly compare crosslinking data to the repressive conformation crystal structure in this current study, we performed XL-MS on apo and T0070907-bound PPARγ LBD in the absence or presence of NCoR ID2 corepressor and TRAP220 ID2 coactivator peptides. In the presence of T0070907, robust crosslinks are detected between K474-K265 (helix 12-Ω loop) and K474-K275 (helix 12-helix 3), which are further enriched in the presence of NCoR ID2 peptide and decreased in the presence of TRAP220 ID2 peptide (Fig. 8b). No active conformation K474-K301 (helix 12-helix 3) crosslinks are detected in the absence or presence of T0070907 when coactivator or corepressor peptide was bound. Although K474-K301 crosslinks were observed for apo-PPARγ in the presence of SRC-1 coactivator ID2 peptide[34], the NCoR ID2 and TRAP220 ID2 peptides we used here bind with much higher affinity (Supplementary Fig. 13), which likely inhibits the K301-K474 crosslink; K301 is part of the charge clamp involved in binding coactivator and corepressor-peptide motifs. Furthermore, active conformation K474-K457 (helix 12-helix 11) crosslinks were dimensioned in the NCoR ID2 peptide-bound form and further decreased in the presence of T0070907. These XL-MS data support the repressive helix 12 conformation within the orthosteric pocket that we captured in the crystal structures of T0070907-bound PPARγ LBD bound to corepressor peptides and validated with PRE NMR.

**Structural mechanism of T0070907-mediated inverse agonism.** We previously solved a crystal structure of T0070907-bound PPARγ LBD without a coregulator peptide where helix 12 did not adopt a repressive conformation but instead showed the crystal contact stabilized active and inactive helix 12 conformations typical of many PPARγ LBD crystal structures solved in the absence of coregulator peptide (e.g., Supplementary Fig. 8a)[29].

Notably, T0070907 adopts different crystallized binding modes in different activity-dependent conformations (Fig. 9a). In the crystal structure solved in the absence of coregulator peptide, which displays an active LBD conformation, the T0070907 pyridyl group occupied the orthosteric pocket, and the nitro group situated near the AF-2 surface. However, in the corepressor-peptide-bound form, which displays a repressive LBD conformation, the T0070907 binding mode changes and the pyridyl group is situated near AF-2 surface allowing helix 12 to occupy the orthosteric pocket. These two different T0070907 binding poses provide a plausible structural basis for the slow conformational exchange ($k_{ex} \gg 2\,s^{-1}$ at room temperature) between two long-lived T0070907-bound conformations we detected using ZZ-exchange NMR[29]. We also found that PPARγ LBD bound to GW9662, a covalent antagonist related to T0070907 that has a phenyl group instead of a pyridyl group, shows only one population of NMR peaks and functions as a transcriptionally neutral antagonist. In that study, we showed that one of the T0070907-bound conformations binds NCoR ID2 peptide with high affinity and the TRAP220 ID2 peptide with weak affinity—a repressive-like conformation. In contrast, the other T0070907-bound conformation, which is shared between T0070907- and GW9662-bound PPARγ, binds the TRAP220 ID2 peptide with high affinity and NCoR ID2 peptide with weak affinity—an active-like conformation.

To gain further insight into the mechanism of action of T0070907, we performed PRE NMR analysis of MTSL-labeled T0070907-bound [K474C]-PPARγ LBD to determine the location of helix 12 in the two slowly exchanging conformations in the absence of coregulator peptide (Fig. 9b). Similar to our previous non-PRE NMR findings[29], a number of residues show a double population of NMR peaks including G399 in the AF-2 surface. Notably, some residues that previously showed only one population of peaks in the absence of PRE spin label now show a double population of peaks in the diamagnetic state, including V248, G344, and G346 on the β-sheet. When considered with our repressive conformation crystal structures (Fig. 2), this indicates that the chemical environment of residues near the β-sheet, which is structurally proximal the helix 12 within the orthosteric ligand-binding pocket, is significantly different after introduction of the MTSL spin label resulting in two structural populations. Indeed, of these residues that show a well resolved double population of peaks, only one of the two NMR peaks shows significant MTSL-induced line broadening in the paramagnetic states (Fig. 9c). These PRE NMR findings are consistent with helix 12 occupying both the repressive conformation (e.g., one of the two peaks for V248, G344, and G346 in the β-sheet are affected) and the active conformation (e.g., one of the two peaks for G399 in the AF-2 surface is affected) when of PPARγ is bound to T0070907.

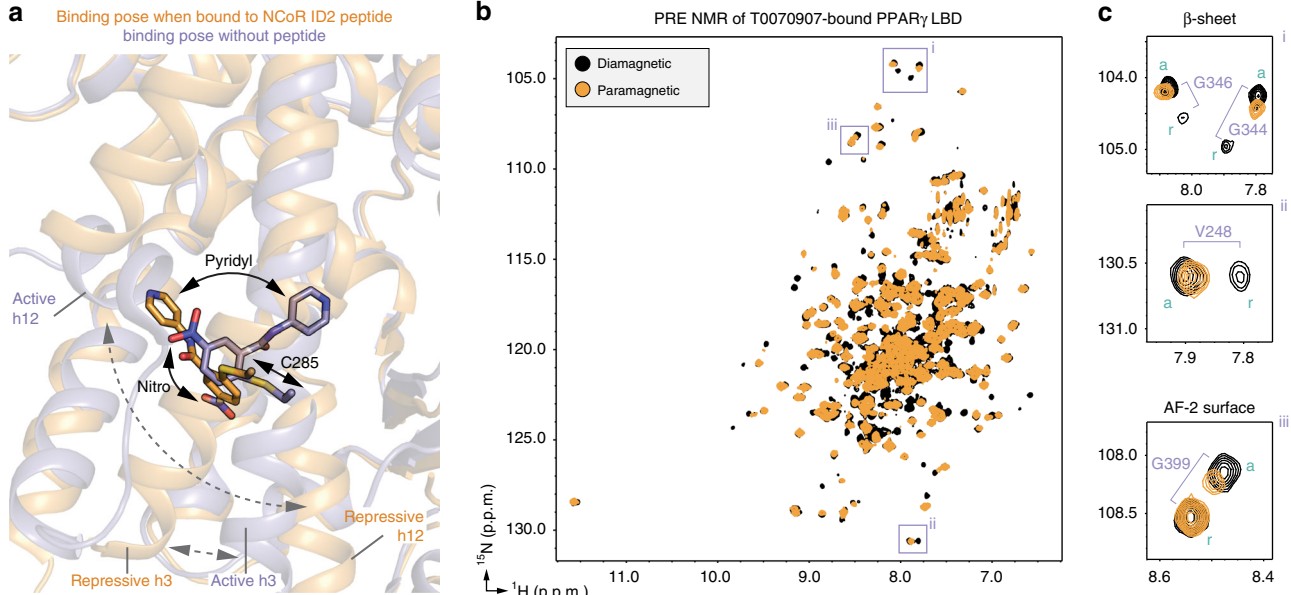

**Fig. 9 T0070907 adopts different activity-dependent binding poses. a** Structural overlay highlighting the T0070907 binding pose in the crystal structure of PPARγ LBD bound to T0070907 and NCoR ID2 peptide (orange; PDB 6ONI) compared with the crystal structure of T0070907-bound PPARγ LBD without a coregulator peptide (blue; PDB 6C1I). **b, c** 2D [$^1$H,$^{15}$N]-TROSY-HSQC spectra from the PRE analysis of MTSL-labeled T0070907-bound [K474C]-PPARγ LBD without coregulator peptide. Purple boxes in (**b**) highlight residues shown the zoomed-in snapshots (**c**) showing that only one of two slowly exchanging populations corresponding to the active (labeled "a") and repressive (labeled "r") is affected by the MTSL spin label on helix 12.

We also performed PRE NMR analysis of MTSL-labeled GW9662-bound [K474C]-PPARγ LBD (Supplementary Fig. 14) to determine how the structural mechanism of GW9662 compares with T0070907. Similar to our previous non-PRE NMR findings[29], most residues show only one population of peaks, including residues in the AF-2 surface and β-sheet—regions that are sensitive to the slow exchange between the two long-lived T0070907-bound active and repressive helix 12 conformations. However, in other structural locations, several residues within or proximal to the β-sheet of MTSL-labeled GW9662-bound PPARγ LBD show notable peak doubling, including K230, V248, and D380. Similar to the T0070907-bound PPARγ PRE data, one of the two NMR peaks is broadened due to spatial proximity to the MTSL spin label in solution. This is consistent with helix 12 occupying the repressive conformation within the orthsteric pocket near the β-sheet when PPARγ is bound to GW9662. Furthermore, for residues showing one NMR peak or population, MTSL-induced line broadening is observed for G399 in the AF-2 surface (i.e., the active helix 12 conformation), as well as G344 and G346 in the β-sheet (i.e., the repressive helix 12 conformation)—the same residues affected in the T0070907 PRE NMR data where helix 12 exchanges between long-lived active and repressive conformations. These data are consistent with our previous $^{19}$F NMR studies showing that helix 12 in GW9662- and T0070907-bound PPARγ exchange between similar conformations with different relative populations[29,30].

Taken together with the apo-PPARγ PRE NMR data (Fig. 7) and our previous findings that revealed a rank order of NCoR ID2 peptide affinity (highest to lowest) of T0070907-bound PPARγ, GW9662-bound PPARγ, and apo-PPARγ[29]—these data suggest that the time scale rate of exchange between an active helix 12 conformation (i.e., solvent exposed) and repressive helix 12 conformation (i.e., within the orthosteric pocket) may influence corepressor selectivity. For apo-PPARγ, the rate of exchange between active and repressive helix 12 conformations is relatively fast (intermediate NMR time scale). Binding of T0070907 slows the rate of exchange considerably, allowing population of a long-

lived helix 12 conformation within the orthosteric ligand-binding pocket that affords high corepressor binding affinity and transcriptional repression. In contrast, GW9662-bound PPARγ likely populates a shorter-lived repressive helix 12 population within the orthosteric pocket that exchanges with an active-like conformation more quickly relative to T0070907-bound PPARγ but slower than apo-PPARγ, resulting in an intermediate NCoR ID2 affinity between T0070907-bound PPARγ and apo-PPARγ.

## Discussion

The studies presented here provide insights into the structural mechanisms of transcription repression of PPARγ and the activation mechanism of agonists. Other reported corepressor-peptide-bound NR crystal structures including ERRγ, FXR, GR, PPARα, PR, RARα, REV-ERBα, RORγ, and RXRα show structurally diverse, solvent-exposed conformations of the AF-2 helix 12 with antagonist or inverse agonist ligands bound to the orthosteric pocket that enable binding of corepressor peptide (Supplementary Fig. 15)[3–14]. In our corepressor-peptide-bound structures of PPARγ, the corepressor-selective inverse agonist T0070907 adopts a binding pose near the AF-2 surface that leaves a large portion of the orthosteric pocket open—double the pocket volume of the coactivator-bound active conformation—which, along with a kink in helix 3 relative to the active conformation, enables helix 12 to bind deep within the pocket near the β-sheet in a repressive conformation. This helix 12 conformation is unique compared with all other published NR crystal structures and enables PPARγ to interact with the ID2 motif of NCoR and SMRT corepressor proteins with high affinity, likely by sequestering helix 12 away from the AF-2 surface where it would physically clash with the bound corepressor ID2 peptide.

To our knowledge, T0070907 is the most efficacious corepressor-selective PPARγ ligand reported to date. However, other PPARγ ligands are capable of maintaining or increasing corepressor selectivity albeit with lesser efficacy, including the covalent antagonist GW9662, noncovalent antagonist SR11023,

and noncovalent inverse agonist SR10171[29,31]. Crystal structures PPARγ LBD bound to these ligands in the absence of coregulator peptides all show solvent exposed, crystal contact-induced helix 12 conformations (Supplementary Fig. 8). Our PRE NMR studies reveal that, similar to T0070907-bound PPARγ, helix 12 in GW9662-bound PPARγ also exchanges between a solvent-exposed conformation and a conformation within the orthosteric ligand-binding pocket. However, the crystallized binding modes of SR11023 and SR10171 within the orthosteric pocket and the associated conformation of helix 3 clash with the repressive helix 12 conformation within the orthosteric pocket (Supplementary Fig. 16). It is possible that SR11023, SR10171, and other related noncovalent inverse agonists including SR2595 and SR10221[36] function through a different helix 12-dependent mechanism. However, the SR10171- and SR11023-bound crystal structures do not report on the conformation of helix 12 in an activity-dependent, corepressor-peptide-bound conformation. Structural studies in the presence of corepressor peptides may provide additional insight into the mechanism of action of these ligands compared with covalent inverse agonists and antagonists (T0070907 and GW9662) that enable helix 12 to exchange between a solvent-exposed conformation and a buried conformation within the orthosteric ligand-binding pocket.

Our previous study identified a water-mediated interaction between the T0070907 pyridyl group and the side chain of R288, and using mutagenesis we found that a positively charged residue (Arg or Lys) is essential for corepressor-selective inverse agonism[29]. When considered with the studies presented here, an extended structural model for PPARγ-corepressor selectivity emerges. R288 is one of several residues that our current mutagenesis identifies as critical for T0070907-mediated corepressor-selective inverse agonism. In the repressive conformation, the binding pose of the T0070907 pyridyl group moves away from R288 by ~90° and points toward the AF-2 surface. Furthermore, an extensive network of side chain interactions contribute to stabilizing helix 12 in the repressive conformation within the orthosteric pocket, including hydrogen bonds between the side chains of R288 on helix 3 to E471 and D475 on helix 12. This suggests there may be two distinct, mutually non-exclusive structural roles for R288: stabilizing the repressive helix 12 conformation within the orthosteric pocket via the T0070907 interaction network (Fig. 6a), and stabilizing T0070907 into an active-like conformation via the pyridyl-water hydrogen bond network[29]—resulting in the two slowly exchanging long-lived T0070907-bound conformations.

Our work here also describes the conformational ensemble of apo-PPARγ helix 12 in solution at atomic resolution. We and others have described that crystal structures of PPARγ LBD in the apo- and ligand-bound states solved in the absence of coregulator peptides look very similar and reveal no overall structural changes that explain the pharmacological activity of ligands except differences in ligand-binding pose[23,25,29,30,34,37,38]. This is likely due to at least two phenomena that may be representative of other NR crystal structures: crystal contacts that influence the conformation of helix 12 in apo- and ligand-bound forms that stabilize an active-like state; and conformational selection for a low energy conformation in the crystals that does not fully represent the dynamic conformational ensemble of helix 12 in solution. Our PRE NMR data show that apo-PPARγ helix 12 exchanges between a transcriptionally active state where helix 12 is solvent exposed and a transcriptionally repressive state within the orthosteric ligand-binding pocket. In a sense, T0070907 could be considered a chemical tool that enables the long-lived stabilization of one of the apo-state helix 12 conformations within the orthosteric pocket. The large conformational changes observed for helix 3, helix 11, and helix 12 between the active and

repressive conformations explain the highly dynamic nature of these regions in apo-PPARγ in previous NMR and HDX-MS structural studies[21,23,25,26,34]. Solution NMR and HDX-MS studies have revealed that other apo-NR LBDs also have a dynamic ligand-binding pocket and AF-2 surface that are stabilized upon agonist binding[39–47]. It is therefore possible that other NRs could share a similar helix 12-dependent mechanism for transcriptional repression, though future studies would need to explore this in detail.

Our data inform a refined model for ligand-induced activation of PPARγ and a potential molecular guide for designing corepressor-selective PPARγ inverse agonists. In the absence of ligand, apo-helix 12 exchanges between a solvent-exposed transcriptionally active conformation and a repressive conformation within the orthosteric ligand-binding pocket (Supplementary Movie 1). Orthosteric ligands compete with the repressive helix 12 conformation for binding to the ligand-binding pocket. Agonist binding displaces or blocks helix 12 from adopting the repressive conformation within the orthosteric pocket and stabilizes a solvent-exposed active helix 12 conformation that enables PPARγ to interact with LXXLL-containing motifs present within transcriptional coactivator proteins with high affinity. Transcriptionally neutral noncovalent orthosteric PPARγ antagonists and repressive noncovalent orthosteric inverse agonists[31,34,36] also compete with the repressive helix 12 conformation for binding to the pocket, but do not stabilize an active helix 12 conformation. Noncovalent orthosteric antagonists may maintain a conformation resulting in a transcriptionally neutral coregulator affinity balance that mimics the coregulator affinity of apo-PPARγ or PPARγ bound to endogenous ligands. On the other hand, noncovalent orthosteric inverse agonists may function by two related mechanisms: competing with endogenous PPARγ ligands that activate PPARγ transcription, and inducing a conformation that weakens coactivator binding without significantly increasing corepressor binding affinity, which in total maintains a relative preference for corepressor binding. In contrast, the corepressor-selective inverse agonist mechanism of T0070907 we describe here is different. Our work suggests an corepressor-selective ligand design model in which ligands that are able to completely displace helix 12 from the AF-2 surface—e.g., by stabilizing helix 12 within the orthosteric pocket—will robustly increase corepressor binding affinity resulting in transcriptional repression of PPARγ.

## Methods

**Materials and reagents**. T0070907 (CAS 313516-66-4), Rosiglitazone (CAS 122320-73-4), MTSL/MTSSL (CAS 81213-52-7), and 3-Maleimido-PROXYL (CAS 5389-27-5) were obtained from Cayman Chemicals, Sigma, or Enzo Life Sciences. Peptides derived from human NCoR ID1 (2044-2066; RTHRLITLADHICQIITQDF ARN), human NCoR ID2 (2256-2278; DPASNLGLEDIIRKALMGSFDDK), human SMRT ID1 (2125-2147; GHQRVVTLAQHISEVITQDYTRH), human SMRT ID2 (2235-2256; TNMGLEAIIRKALMGKYDQWEE), human TRAP220 ID1 (residues 597–615; KVSQNPILTSLLQITGNGG), and human TRAP220 ID2 (residues 638–656; NTKNHPMLM NLLKDNPAQD) were synthesized by LifeTein with an amidated C-terminus for stability, with or without a N-terminal FITC label and a six-carbon linker (Ahx). PPARγ LBD mutant variants were created from the plasmids listed below using site-directed mutagenesis with primers listed in Supplementary Table 3.

**Protein expression, purification, and characterization**. Wild type or mutant human PPARγ ligand-binding domain (LBD) protein, residues 203–477 (isoform 1 numbering), was expressed from a pET46 Ek/LIC plasmid (Novagen) as a TEV-clevable N-terminal hexahistag fusion protein in *Escherichia coli* BL21(DE3) cells using autoinduction ZY media (unlabeled protein), or using M9 minimal media (for NMR studies) supplemented with $^{15}N$ ammonium chloride with or without $^{13}C$-glucose and $D_2O$ at 37 °C. For M9 growth, cells were induced with 1.0 mM isopropyl β-D-thiogalactoside (M9) at an $OD_{600nm}$ of 0.6, grown for an additional 24–48 h at 18 °C then harvested. For ZY growth, cells were grown for 5 h at 37 °C and additional 12–18 h at 22 °C then harvested. Cells were lysed and 6xHis-PPARγ LBD was purified using Ni-NTA affinity chromatography and gel filtration

chromatography. TEV protease was used to cleave the histag for most experiments except protein used for TR-FRET and fluorescence polarization. The purified proteins were concentrated to 10 mg mL$^{-1}$ in a buffer consisting of 20 mM potassium phosphate (pH 7.4), 50 mM potassium chloride, 5 mM TCEP, and 0.5 mM EDTA (phosphate buffer). Purified protein was verified by SDS-PAGE as >95% pure. Covalent binding of T0070907 to PPARγ LBD (wild type or mutant variants) was analyzed by ESI-MS using a LTQ XL linear Ion trap mass spectrometer (Thermo Scientific); samples were incubated with or without 2 molar equivalents of T0070907 (unless otherwise indicated below) at 4 °C overnight and diluted to 2–3 µM in 0.1% formic acid for ESI-MS analysis.

**Cellular two-hybrid protein–protein interaction assay.** HEK293T (ATCC CRL03216) were cultured in Dulbecco's minimal essential medium (DMEM, Gibco) supplemented with 10% fetal bovine serum (FBS) and 50 units ml$^{-1}$ of penicillin, streptomycin, and glutamine. Cells were plated 20,000 cells/well in a 96-well flat bottom cell culture plate and co-transfected with 100 ng pG5-UAS and 25 ng pCMV-Gal4-PPARγ (human residues 185–477, isoform 1 numbering, containing the hinge and LBD) along with pACT empty vector (Promega) expressing the VP16 transactivation domain only, pACT with VP16 fused to the mouse NCoR receptor interaction domain (RID, residues 1828–2471), or pAct with VP16 fused to the NCoR RID with each critical residue of the ID2 motif (LEDIIRKAL) mutated to threonine (TEDTTRKAT). Transfection solutions were prepared in Opti-MEM with Mirus Bio TransIT-LT1 transfection reagent. After 24 h incubation at 37 °C in a 5% CO$_2$ incubator, DMSO or T0070907 was added at a final concentration of 0.01% or 10 µM, respectively. After another 24 hr incubation, britelite plus (PerkinElmer) was added to each well, mixed, then transferred to a white-bottom 384-well plate and read on a BioTek Synergy Neo multimode plate reader. Data were plotted and analyzed using GraphPad Prism software.

**Cellular transcriptional reporter assay.** HEK293T (ATCC CRL03216) were cultured in Dulbecco's minimal essential medium (DMEM, Gibco) supplemented with 10% fetal bovine serum (FBS) and 50 units ml$^{-1}$ of penicillin, streptomycin, and glutamine. Cells were grown to 90% confluency in T-75 flasks; from this, 2 million cells were seeded in a 10-cm cell culture dish for transfection using X-tremegene 9 (Roche) and Opti-MEM (Gibco) with full-length human PPARγ (isoform 2) expression plasmid (4 µg), and a luciferase reporter plasmid containing the three copies of the PPAR-binding DNA response element (PPRE) sequence (3xPPRE-luciferase) (4 µg). After an 18-h incubation, cells were transferred to white 384-well cell culture plates (Thermo Fisher Scientific) at 10,000 cells/well in 20 µL total volume/well. After a 4 h incubation, cells were treated in quadruplicate with 20 µL of either vehicle control (1.5% DMSO in DMEM media), twofold serial dilution of TZDs for dose response experiments, or 5 µM ligand. After a final 18-h incubation, cells were harvested with 20 µL Britelite Plus (PerkinElmer), and luminescence was measured on a BioTek Synergy Neo multimode plate reader. Data were plotted in GraphPad Prism as luminescence vs. ligand concentration and fit to a sigmoidal dose response curve.

**TR-FRET biochemical assays.** Time-resolved fluorescence resonance energy transfer (TR-FRET) assays were performed in low-volume black 384-well plates (Greiner) using 23 µL final well volume. For the TR-FRET coregulator peptide interaction assay, each well contained 4 nM protein (WT or mutant 6xHis-PPARγ LBD), 1 nM LanthaScreen Elite Tb-anti-His Antibody (ThermoFisher #PV5895), and 400 FITC-labeled TRAP220, NCoR and SMRT peptide in TR-FRET buffer (20 mM potassium phosphoate, 50 mM KCl, 5 mM TCEP, 0.005% Tween 20, pH 7.4). For the ligand displacement assay, each well contained 1 nM protein (wild type or mutant 6xHis-PPARγ LBD), 1 nM LanthaScreen Elite Tb-anti-HIS Antibody (ThermoFisher #PV5895), and 5 nM Fluormone Pan-PPAR Green (Invitrogen; #PV4894) in TR-FRET buffer at pH 8. For all assays, compounds stocks were prepared via serial dilution in DMSO, added to wells in triplicate, and plates were read using BioTek Synergy Neo multimode plate reader at 25 °C for several time points between 1 and 24 h. The Tb donor was excited at 340 nm, its emission was measured at 495 nm, and the acceptor FITC emission was measured at 520 nm. Data were plotted using GraphPad Prism as TR-FRET ratio (520 nm/495 nm) vs. ligand concentration. The coregulator interaction data were fit to sigmoidal dose response curve equation, and ligand displacement data were fit to the one site–Fit K$_i$ binding equation using the binding affinity of Fluormone Pan-PPAR Green (2.8 nM; Invitrogen #PV4894 product insert).

**Fluorescence polarization assays.** Wild type or mutant 6xHis-PPARγ LBD (with or without T0070907) was serially diluted in assay buffer (20 mM potassium phosphate, 50 mM potassium chloride, 5 mM TCEP, 0.5 mM EDTA, 0.01% Tween-20, pH 8) and plated with 180 nM FITC-labeled TRAP220 ID2 or NCoR ID2 peptides in low-volume black 384-well plates (Greiner) in triplicate. For single PPARγ LBD concentration experiments, the final concentration of FITC-labeled SMRT (ID1 and ID2), NCoR (ID1 and ID2), and TRAP220 ID2 peptides was 180 nM and wild-type 6xHis-PPARγ LBD concentration was 25 µM. Plates were incubated at 25 °C for 1 h, and fluorescence polarization was measured on a BioTek Synergy Neo multimode plate reader at 485 nm emission and 528 nm excitation wavelengths. Data were plotted using GraphPad Prism as fluorescence polarization

signal in millipolarization units vs. protein concentration and fit to a one site—total binding equation.

**X-ray crystallography and structure refinement.** Purified PPARγ LBD was incubated with rosiglitazone or T0070907 at a 1:3 protein/ligand molar ratio in PBS overnight, then incubated with corresponding peptides at a 1:5 protein/peptide molar ratio and concentrated to 10 mg/mL and buffer exchanged into phosphate buffer to remove DMSO and unbound ligands and peptides. Protein complex crystals were obtained after 5–8 days at 22 °C by sitting-drop vapor diffusion against 50 µL of well solution using 96-well format crystallization plates. The crystallization drops contained 1 µL of protein complex sample mixed with 1 µL of reservoir solution containing 0.1 M Tris (pH 8.5), 0.2 M sodium acetate trihydrate, and 30% (w/v) PEG 4000 for rosiglitazone/TRAP220 ID2-bound PPARγ LBD complex; and 0.1 M MES (pH 6.5), 0.2 M ammonium sulfate, and 30% (w/v) PEG 8000 for T0070907/NCoR ID2-bound and T0070907/SMRT ID2-bound PPARγ LBD complexes. Crystals were flash-frozen in liquid nitrogen before data collection. Data collection was carried out at SSRL Beamline 12-2 (SLAC National Accelerator Laboratory). Data were processed, integrated, and scaled with the programs Mosflm and Scala in CCP4[48,49]. The structure was solved by molecular replacement using the program Phaser[50] implemented in the PHENIX package[51] using a previously published PPARγ LBD structure (PDB code: 1PRG)[16] as the search model. The structure was refined using PHENIX with several cycles of interactive model rebuilding in Coot[52]. Pocket volume analysis was performed using CASTp[53]. The Ramachandran statistics are as follows: PPARγ LBD bound to Rosiglitazone and TRAP220 ID2 peptide, 96.95% favored, 0.38% outliers; PPARγ LBD bound to T0070907 and NCoR ID2 peptide, 97.76% favored, 0.00% outliers; PPARγ LBD bound to T0070907 and SMRT ID2 peptide, 96.98% favored, 0.00% outliers. The structural extrapolation between the active conformation (PDB 6ONJ) and repressive conformation (PDB 6ONI) was performed using the morph conformations plug-in within Chimera[54]. To visualize the location of MTSL for the PRE NMR studies, residue K474 was first modeled into cysteine (C474), then the MTSL ligand was imported into Coot as a SMILES string: CC1(C)C(CSS(C)(=O)=O) =CC(C)(C)N1[O]. The MTSL sulfinic acid leaving group was removed and placed adjacent to C474, after which it was regularized and covalently attached to C474.

**NMR spectroscopy.** NMR experiments were acquired at 298 K on a Bruker 700 MHz NMR instrument equipped with a QCI cryoprobe in NMR buffer (50 mM potassium phosphate, 20 mM potassium chloride, 1 mM TCEP, pH 7.4, 10% D$_2$O). Samples for 2D NMR analysis, including PRE NMR, were performed using 200–350 µM $^{15}$N-labeled protein (wild type, K474C, and C285S/K474C PPARγ LBD). Samples bound to T0070907 were incubated with 2 molar equivalents overnight at 4 °C followed by buffer exchange; samples bound to rosiglitazone were added at 2 molar equivalents. Ligand-bound samples were then incubated with 2 molar equivalents of NCoR or TRAP220 peptide. Samples for PRE NMR experiments were incubated with 10 molar equivalents of MTSL (Enzo Life Sciences #ALX-430-134-M050) or 3-Maleimido-PROXYL (Sigma #253375) spin labels overnight at 4 °C followed by buffer exchange; diamagnetic samples were prepared by adding 5× molar excess of fresh sodium ascorbate to reduce the spin label. Of note, because reducing the MTSL from the paramagnetic (nitroxide) to diagmagnetic (hydroxylamine) state changes the protonation state of the MTSL spin label, protein NMR chemical shift changes are often observed in the paramagnetic and diamagnetic forms. [$^2$H,$^{13}$C,$^{15}$N]-labeled PPARγ LBD bound to T0070907 and NCoR ID2 peptide (1 mM) was used to collect 3D NMR experiments for chemical shift assignment, including TROSY-based HNCO, HN(CA)CO, HNCA, HN(CO) CA, HN(CA)CB, HN(COCA)CB, and [$^1$H,$^{15}$N]-NOESY-HSQC data. Several additional 3D NMR datasets were collected to facilitate transfer of NMR chemical shift assignments from T0070907/NCoR ID2-bound PPARγ LBD (BMRB entry 50000) and rosiglitazone-bound PPARγ LBD (BMRB entry 17975)[25] to the PRE NMR samples using the minimum chemical shift perturbation procedure[55]: 3D TROSY-based HNCO and HNCA using a 600 µM [$^{13}$C,$^{15}$N]-labeled apo-PPARγ LBD, and 3D [$^1$H,$^{15}$N]-NOESY-HSQC data using the reduced MTSL-labeled PRE NMR samples. Data were processed and analyzed using Topspin 3.2 or 4.0 (Bruker Biospin), NMRFx processor[56], and NMRViewJ[57].

**BS3 chemical crosslinking mass spectrometry.** Experiments were performed using methods we previously described for PPARγ LBD BS3 XL-MS studies with or without ligand and coregulator peptides[34]. PPARγ LBD (10 µM) with or without T0070907 (10 molar equivalents), NCoR ID2 peptide (5 molar equivalents), or TRAP220 ID2 peptide (5 molar equivalents) were incubated with 50-fold molar excess of BS3 (ThermoFisher #21580) for 1 h. BS3 crosslinking reactions were quenched using 50 mM Tris buffer. Protein samples were sequentially overnight digested by trypsin (trypsin: protein ratio = 1:20, w/w) (Promega #V5111) first and then chymotrypsin (chymotrypsin: protein ratio = 1:20, w/w) (Promega #V1061) prior to analysis by LC-MS/MS. Chymotrypsin was used to introduce a specific cleavage between T473 and K474 for identification of a shorter peptide K474-Y477 at the extreme C-terminus of the AF-2 helix 12. Plink2 software was used to analyze and identify crosslinked peptides[58]. The parameters for pLink2 search were as follows: three missed cleavage sites for trypsin/chymotrypsin for each chain; peptide length 4–100 amino acid; pLink2 search results were filtered by requiring

precursor tolerance (±10 p.p.m.) and fragment tolerance (±15 p.p.m.) and FDR below 5% was required for all identified XL-MS peaks. The ion intensities and peak areas of the crosslinked peptides from different experimental conditions were manually calculated and compared. The highest peak intensity across all indicated experimental conditions within each bar plot panel was normalized as 100%—first to raw peak area of a control peptide lacking a lysine residue (S289-VEAVQEITE-Y299) to normalize for sample injection, then to the highest mean peak area within each condition. Experiments were performed in triplicate.

**Reporting summary**. Further information on research design is available in the Nature Research Reporting Summary linked to this article.

## Data availability

Crystal structures generated during the current study are available in the Protein Data Bank (PDB) under accession codes 6ONJ (PPARγ LBD bound to rosiglitazone and TRAP220 ID2 peptide), 6ONI (PPARγ LBD bound to T0070907 and NCoR ID2 peptide), and 6PDZ (PPARγ LBD bound to T0070907 and SMRT ID2 peptide). NMR chemical shift assignments generated during the current study of PPARγ LBD bound to T0070907 and NCoR ID2 peptide are available in the Biological Magnetic Resonance Data Bank (BMRB) under entry ID 50000. The source data underlying the Fig. 1a–d, 5c–d, 6c–i, 7c, 8b and Supplementary Figs. 4, 5, 6, 7, and 13 are provided in a separate Source Data file. All other datasets generated and/or analyzed during the current study are available from the corresponding author on reasonable request.

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

## Acknowledgements

We thank Paola Munoz-Tello and Ted Kamenecka for helpful discussions and input. This work was supported by National Institutes of Health (NIH) grants F32DK108442 (R.B.), R01DK105825 (P.R.G.), and R01DK101871 (D.J.K.); Richard and Helen DeVos graduate fellowship award (S.M.); and National Science Foundation (NSF) award 1359369 that funds the SURF program at Scripps Research Florida (S.M.). Use of the Stanford Synchrotron Radiation Lightsource, SLAC National Accelerator Laboratory, is supported by the U.S. Department of Energy, Office of Science, Office of Basic Energy Sciences under Contract No. DE-AC02-76SF00515. The SSRL Structural Molecular Biology Program is supported by the DOE Office of Biological and Environmental Research, and by the National Institutes of Health, National Institute of General Medical Sciences (including P41GM103393). The contents of this publication are solely the responsibility of the authors and do not necessarily represent the official views of NIDDK, NIGMS, or NIH.

## Author contributions

J.S. and D.K. conceived and designed the research. J.S. and J.B. expressed and purified protein. J.S. solved the crystal structures and performed NMR and biochemical assays. S.M. performed the mammalian two-hybrid assay; A.N. and L.S. generated the co-repressor RID plasmids. J.S., R.B., and J.B. performed the transcription assays. J.Z. and P.R.G. performed chemical crosslinking mass spectrometry experiments. D.K. supervised the research and wrote the manuscript along with J.S. and input from all authors who approved the final version.

## Competing interests

The authors declare no competing interests.
