## [Peer Review File · Nature Communications]

Reviewers' comments:

Reviewer #1 (Remarks to the Author):

The activity of nuclear receptor (NR) is regulated by the recruitment of cofactor (coactivator or corepressor) in a ligand-inducible manner. While it is well-established that agonist activates NR by inducing the AF-2/helix 12 of NR to adopt an active conformation that facilitates the binding of coactivator via the LXXLL motif, the mechanism responsible for how inverse agonist represses NR through corepressor recruitment is less unified. In addition, the hypothesis that the helix-12 of Apo-NR LBD can exchange between active and repressive conformations has not been fully investigated and defined. In this manuscript, the authors use PPAR γ as a model receptor to investigate these outstanding problems in the NR field, by first solving the co-crystal structures of PPAR γ , an inverse agonist (T0070907), and corepressor peptides. The crystal structures reveal a unique corepressor-bound transcriptionally repressive conformation not previously reported, in which the AF-2/helix -12 is displaced from the active conformation and occupies the ligand-binding pocket to double its volume. The unique conformation was further validated by an NMR approach, which also reveals that Apo-helix 12 exchanges between active and repressive states. These data provide additional mechanistic insights into the different cellular outcomes (active, neutral and repressive) induced by ligands of different chemical properties through recruitment of cofactors. This new information is of interest to readers in the nuclear receptor research community, especially those interested in the structural basis of ligand activity.

Additional data, including those from cell-based assays will strengthen the prediction made from the structural observations, and correlate inverse agonist-induced "transcriptional repressive conformation" to "cellular transcriptional repression by an inverse agonist". In addition, fixing a few minor errors will improve the overall quality of the manuscript.

Major points:

1. Fig 1A: in the two-hybrid system, is the NCoR ID2 alone sufficient to interact with PPAR γ LBD?
2. Fig 4c and 4d: FP assay was used to measure the interaction between corepressor peptide and PPAR γ (WT vs. mutant) in the absence of an inverse agonist. Since inverse agonist-induced PPAR γ – corepressor interaction is more relevant to the transcriptionally repressive conformation, the FP assays should also be performed in the presence of an inverse agonist.
3. To correlate structural conformation with functional outcomes, it is critical to test all the PPAR γ mutants tested in FP (Fig 4c and 4d) and TR-FRET (Fig 5) assays in a cell-based transcriptional reporter assay.
4. At the end of the discussion, the authors stated that "These findings reveal the structural basis for a molecular switch regulating transcriptional repression and activation of PPAR γ and provide a molecular guide for designing corepressor-selective PPAR γ inverse agonists." Please elaborate more on how to develop a more potent inverse agonist based on T0070907. It appears that T0070907 is capable of expanding the volume of the ligand-binding pocket, and enhancing the corepressor's binding affinity to PPAR γ .

Minor points:

1. Line 43: please fix this sentence: "hypothesized thought to...".
2. Line 47: please fix this sentence "some of which some are...".
3. Fig 5: please label each panel (a – f) correctly (e.g., the second "d" should be "e"; the current "e" should be "f").

Reviewer #2 (Remarks to the Author):

In this report, the authors have determined crystal structures of peroxisome proliferator-activated receptor gamma (PPAR γ) bound to an inverse agonist (T0070907) and corepressor peptides (NCoR and SMRT), and that bound to an agonist (Rosiglitazone) and an activator peptide (TRAP220). In the structure of T0070907-bound PPAR γ in complex with corepressor peptides, the helix 12 is located in the ligand-binding pocket, unlike any other PPAR γ structures. The interactions observed

in the structures were confirmed by mutagenesis studies. Furthermore, the location of the helix 12 of PPAR γ in repressive and active conformations as well as apo-PPAR γ were assessed by PRE NMR studies.

The comments of this reviewer are as follows.

1. PRE distance information obtained from one spin label is ambiguous because of the nature of MTSL side chain, which is basically flexible causing large error, and sometime fixed in a certain orientation leading a biased result. So, it is preferable to use two or more different spin-labeled samples in order to obtain information of higher accuracy.
2. In order to show an effect of the MTSL spin label itself on the protein structure, please show an overlay of TROSY spectra of the diamagnetic MTSL-attached protein and the intact protein.
3. In the previous report (Brust et al., Nature Communications, 2018), the authors showed that PPAR γ exchanges between two long-lived conformations when bound to T0070907, and one of those conformations is similar to the corepressor-bound state, which enables the strong binding of PPAR γ to corepressor peptides. While in this report, the authors showed that apo-PPAR γ helix 12 also exchanges between two conformations, one of which is similar to repressive state. To clarify the structural mechanism of the function of T0070907, please discuss the difference of the structures of apo-PPAR γ and T0070907-bound PPAR γ , and a reason of the higher affinity of T0070907-bound PPAR γ to corepressor peptides.

Reviewer #3 (Remarks to the Author):

In this manuscript, the authors report the crystal structures of 2 complexes of PPAR γ with T0070907 inverse agonist and corepressor peptides as well as the crystal structure of PPAR γ in complex with rosiglitazone and Trap220 coactivator. The structures of the corepressor complexes reveal a new repressive conformation of the receptor. The authors also describe paramagnetic relaxation NMR spectroscopy analysis that agrees with the proposed exchange mechanism between active and repressive conformations of the receptor.

Major comments

1. The new repressive conformation of the complex with corepressor peptides and inverse agonist result from the insertion of helix 12 into the pocket and the shifts of helix 2b and the strand b1. A previously reported structure of PPAR γ with SR11023, although obtained without corepressor peptide, revealed a conformation of helix 12 closed to the N-terminal part of H3 and H2' and also compatible with chemical crosslinking mass spec data. This structure should be compared.
2. Kojetin and colleagues previously published in Nature Communication a study on the comparison of the PPAR γ complexes with the inverse agonist T0070907 and the antagonist GW9662 revealing the role of R288 and a water molecule that were proposed to explain the difference of activity of the 2 compounds. The interactions of R288 in the present study should be discussed as well as the expected difference in GW9662 induced PPAR γ conformation and activity and supported by data (effect of mutations,...).
3. The authors mention that the repressive helix 12 conformation is not influenced by the crystal packing as shown on Fig. S8. The figure should be clarified. In addition the density of the loop connecting H11 to H12 should also be shown.
4. To validate the functional importance of the repressive conformation, the authors mutated various AA and analyze their effect on repressor interactions. They should also show that the PPAR γ mutations prevent corepressor in the context of full-length proteins by mammalian 2-hybrid assay for example.

Minor comments

-Page 2. Line 47: "...some of which some..."

-Page 4. Line 101-109. The active conformation of PPAR γ is already known. This paragraph should be shortened.

-Page 9, line 256-257: This sentence is unclear.

-Crystal structures validation:

The validation reports suggest that following structures can be improved: 6ONI (Rfree, Ramachandran outliers), 6ONJ (side chain outliers, Ramachandran outliers), and 6PDZ (Ramachandran outliers)

We thank the reviewers for their comments and suggestions regarding our manuscript and the editor for overseeing the review process. We have addressed the concerns by revising and updating the manuscript and Supplementary Information documents based on the reviewer suggestions. We include a related manuscript file (PDF file) that highlights the changes we made to the main manuscript using Microsoft Word “Track Changes” format; changes made in the Supplementary Information are not highlighted. Our revised manuscript complies with the *Nature Communications* editorial requests and formatting guidelines. Below we provide a point-by-point response to the reviewer critiques.

Reviewer Comments

Reviewer #1

The activity of nuclear receptor (NR) is regulated by the recruitment of cofactor (coactivator or corepressor) in a ligand-inducible manner. While it is well-established that agonist activates NR by inducing the AF-2/helix 12 of NR to adopt an active conformation that facilitates the binding of coactivator via the LXXLL motif, the mechanism responsible for how inverse agonist represses NR through corepressor recruitment is less unified. In addition, the hypothesis that the helix-12 of Apo-NR LBD can exchange between active and repressive conformations has not been fully investigated and defined. In this manuscript, the authors use PPAR γ as a model receptor to investigate these outstanding problems in the NR field, by first solving the co-crystal structures of PPAR γ , an inverse agonist (T0070907), and corepressor peptides. The crystal structures reveal a unique corepressor-bound transcriptionally repressive conformation not previously reported, in which the AF-2/helix -12 is displaced from the active conformation and occupies the ligand-binding pocket to double its volume. The unique conformation was further validated by an NMR approach, which also reveals that Apo-helix 12 exchanges between active and repressive states. These data provide additional mechanistic insights into the different cellular outcomes (active, neutral and repressive) induced by ligands of different chemical properties through recruitment of cofactors. These new information is of interest to readers in the nuclear receptor research community, especially those interested in the structural basis of ligand activity.

Additional data, including those from cell-based assays will strengthen the prediction made from the structural observations, and correlate inverse agonist-induced “transcriptional repressive conformation” to “cellular transcriptional repression by an inverse agonist”. In addition, fixing a few minor errors will improve the overall quality of the manuscript.

Major points:

1. Fig 1A: in the two-hybrid system, is the NCoR ID2 alone sufficient to interact with PPAR γ LBD?

Authors’ response: Yes, our mammalian two-hybrid data (now shown in **Fig. 1b**) indicate that the ID2 motif is sufficient to interact with the PPAR γ LBD. To illustrate this better to readers, we created a new Results subsection titled “*The corepressor ID2 motif is sufficient for PPAR γ interaction*” containing the functional data that support this idea. We separated these functional data (now **Fig. 1**) from the crystal structures (now **Fig. 2**), and for clarity and we added a previous SI figure (was Fig. S1) to **Fig. 1** showing that the corepressor ID2 peptides robustly interact with apo-PPAR γ LBD but the ID1 peptides do not.

2. Fig 4c and 4d: FP assay was used to measure the interaction between corepressor peptide and PPAR γ (WT vs. mutant) in the absence of an inverse agonist. Since inverse agonist-induced PPAR γ – corepressor interaction is more relevant to the transcriptionally repressive conformation, the FP assays should also be performed in the presence of an inverse agonist.

Authors’ response: As requested by the reviewer, we performed FP assays for T0070907-bound PPAR γ LBD. These new data are now shown in **Fig. 5c** and **Supplementary Fig. 4** along with the FP data of apo-PPAR γ and new cellular transcriptionally assay data.

3. To correlate structural conformation with functional outcomes, it is critical to test all the PPAR γ mutants tested in FP (Fig 4c and 4d) and TR-FRET (Fig 5) assays in a cell-based transcriptional reporter assay.

Authors' response: As requested by the reviewer, we tested the mutants in a cell-based transcriptional reporter assay using full-length PPAR γ and a 3xPPRE-luciferase reporter plasmid. These new data are shown in **Figs. 5d and 6i** along with the new T0070907-bound FP data. Overall, there is good agreement between the FP assay (corepressor peptide affinity) and cell-based transcriptional reporter assay findings.

4. At the end of the discussion, the authors stated that “These findings reveal the structural basis for a molecular switch regulating transcriptional repression and activation of PPAR γ and provide a molecular guide for designing corepressor-selective PPAR γ inverse agonists.” Please elaborate more on how to develop a more potent inverse agonist based on T0070907. It appears that T0070907 is capable of expanding the volume of the ligand-binding pocket, and enhancing the corepressor's binding affinity to PPAR γ .

Authors' response: We revised the final discussion paragraph to better differentiate the corepressor-selective inverse agonism mechanism of T0070907 compared to other pharmacological PPAR γ ligands. In short, our data suggest that ligands that are able to stabilize helix 12 within the orthosteric pocket may increase corepressor binding affinity and provide robust corepressor-selective inverse agonism of PPAR γ . The expanded pocket volume in the crystal structure of PPAR γ LBD bound to T0070907 and corepressor peptide is likely representative of apo-PPAR γ as well based on our PRE NMR data of apo-PPAR γ that indicate helix 12 exchanges between a solvent exposed “active” conformation and a buried conformation within the orthosteric pocket.

Minor points:

1. Line 43: please fix this sentence: “hypothesized thought to...”.
2. Line 47: please fix this sentence “some of which some are...”.
3. Fig 5: please label each panel (a – f) correctly (e.g., the second “d” should be “e”; the current “e” should be “f”).

Authors' response: We fixed the errors listed above in minor points 1–3.

Reviewer #2

In this report, the authors have determined crystal structures of peroxisome proliferator-activated receptor gamma (PPAR γ) bound to an inverse agonist (T0070907) and corepressor peptides (NCoR and SMRT), and that bound to an agonist (Rosiglitazone) and an activator peptide (TRAP220). In the structure of T0070907-bound PPAR γ in complex with corepressor peptides, the helix 12 is located in the ligand-binding pocket, unlike any other PPAR γ structures. The interactions observed in the structures were confirmed by mutagenesis studies. Furthermore, the location of the helix 12 of PPAR γ in repressive and active conformations as well as apo-PPAR γ were assessed by PRE NMR studies.

The comments of this reviewer are as follows.

1. PRE distance information obtained from one spin label is ambiguous because of the nature of MTSL side chain, which is basically flexible causing large error, and sometime fixed in a certain orientation leading a biased result. So, it is preferable to use two or more different spin-labeled samples in order to obtain information of higher accuracy.

Authors' response: We use PRE NMR only to obtain a qualitative, general location of helix 12 in solution. PRE distance restraints used in an NMR-based structure calculation may require a more rigorous analysis; however, we searched the literature and nearly all PRE NMR studies we found use a single spin label (typically MTSL). However, in response to this comment, we collected new PRE NMR data using the 3-Maleimido-PROXYL spin label for each of the conditions we studied in **Fig. 7**: the repressive conformation

(bound to T0070907 and NCoR ID2 corepressor peptide), the active conformation (bound to rosiglitazone and TRAP220 ID2 coactivator peptide), and apo-PPAR γ LBD. Overall there is good qualitative agreement between these new 3-Maleimido-PROXYL PRE NMR data (**Supplementary Fig. 10**) and the MTSL PRE NMR data (now **Fig. 7**).

2. In order to show an effect of the MTSL spin label itself on the protein structure, please show an overlay of TROSY spectra of the diamagnetic MTSL-attached protein and the intact protein.

Authors' response: These new spectral overlays are now shown in **Supplementary Fig. 11**, which show relatively minor chemical shift changes—this is expected because the reduced MTSL group will change the chemical environment for NMR peaks corresponding to residues within close proximity to the spin label. These data indicate there are no large structural changes caused by attachment of the spin label.

3. In the previous report (Brust et al., Nature Communications, 2018), the authors showed that PPAR γ exchanges between two long-lived conformations when bound to T0070907, and one of those conformations is similar to the corepressor-bound state, which enables the strong binding of PPAR γ to corepressor peptides. While in this report, the authors showed that apo-PPAR γ helix 12 also exchanges between two conformations, one of which is similar to repressive state. To clarify the structural mechanism of the function of T0070907, please discuss the difference of the strictures of apo-PPAR γ and T0070907-bound PPAR γ , and a reason of the higher affinity of T0070907-bound PPAR γ to corepressor peptides.

Authors' response: In response to this comment, we collected new PRE NMR data on PPAR γ bound to T0070907 (**Fig. 9**)—and in response to a comment from Reviewer 3, GW9662 (**Supplementary Fig. 14**)—in the absence of coregulator peptide to compare to the apo-PPAR γ PRE NMR data. These data are described in the new results section “*Structural mechanism of T0070907-mediated inverse agonism*” along with a comparison to our previous report (Brust et al., Nature Communications, 2018). In short, in this new results section we describe how our data suggest that the time scale rate of exchange between an active (i.e., solvent exposed) and repressive (i.e., within the orthosteric pocket) helix 12 conformation may influence corepressor selectivity.

Reviewer #3

Comment

In this manuscript, the authors report the crystal structures of 2 complexes of PPAR γ with T0070907 inverse agonist and corepressor peptides as well as the crystal structure of PPAR γ in complex with rosiglitazone and Trap220 coactivator. The structures of the corepressor complexes reveal a new repressive conformation of the receptor. The authors also describe paramagnetic relaxation NMR spectroscopy analysis that agrees with the proposed exchange mechanism between active and repressive conformations of the receptor.

Major comments

1. The new repressive conformation of the complex with corepressor peptides and inverse agonist result from the insertion of helix 12 into the pocket and the shifts of helix 2b and the strand b1. A previously reported structure of PPAR γ with SR11023, although obtained without corepressor peptide, revealed a conformation of helix 12 closed to the N-terminal part of H3 and H2' and also compatible with chemical crosslinking mass spec data. This structure should be compared.

Authors' response: As requested, in the new results section “*Validation with chemical crosslinking mass spectrometry data*” we now compare PPAR γ crystal structures bound to SR11023 and SR10171 (solved without coregulator peptide) to our structural findings and our previous chemical crosslinking mass spectrometry (XL-MS) study performed using coactivator and corepressor peptides and several different PPAR γ ligands (Zheng et al, 2018). Because this previous XL-MS study did not study T0070907, we also

collected new XL-MS data \pm T0070907 (\pm the NCoR and TRAP220 ID2 peptides we used in this current study), which are now shown in **Fig. 8**. In short, our repressive conformation crystal structure (bound to T0070907 and corepressor peptide) is compatible with two critical crosslinks in the repressive conformation (K474-K265 and K474-K275), which show distances of 15.6Å and 9.6Å, respectively. Both of these crosslinks are simultaneously compatible with the BS3 crosslinker distance of 11.3Å assuming a modest degree of structural flexibility (e.g., 3–6Å) in solution (Merkley et al., 2014). However, the SR11023-bound helix 12 conformation (without peptide) does not seem to be as compatible—the K474-K265 and K474-K275 distances are much longer, 24.5Å and 28Å, respectively. The distances in the SR10171 structures are similarly much longer (~22Å each). Note that we had to model the lysine side chains for K265 and K275 in the SR11023 and SR10171 crystal structures using the mutagenesis command in PyMOL since they were not modeled in previously likely due to weak electron density in these regions.

2. Kojetin and colleagues previously published in Nature Communication a study on the comparison of the PPAR γ complexes with the inverse agonist T0070907 and the antagonist GW9662 revealing the role of R288 and a water molecule that were proposed to explain the difference of activity of the 2 compounds. The interactions of R288 in the present study should be discussed as well as the expected difference in GW9662 induced PPAR γ conformation and activity and supported by data (effect of mutations,...).

Authors' response: In response to this comment, we collected new PRE NMR data on PPAR γ bound to T0070907 (**Fig. 9**)—and in response to a comment from Reviewer 3, GW9662 (**Supplementary Fig. 14**)—in the absence of coregulator peptide to compare to the apo-PPAR γ PRE NMR data. These data are described in the new results section “*Structural mechanism of T0070907-mediated inverse agonism*” along with a comparison to our previous report (Brust et al., Nature Communications, 2018). In short, in this new results section we describe how our data suggest that the time scale rate of exchange between an active (i.e., solvent exposed) and repressive (i.e., within the orthosteric pocket) helix 12 conformation may influence corepressor selectivity.

Related to the role of R288, we added a new paragraph to the discussion that describes two distinct, mutually non-exclusive structural roles for R288: stabilizing the repressive helix 12 conformation, and stabilizing T0070907 into an active-like conformation that in combination with the repressive T0070907 interaction network results in the two slowly exchanging long-lived T0070907-bound conformations.

3. The authors mention that the repressive helix 12 conformation is not influenced by the crystal packing as shown on Fig. S8. The figure should be clarified. In addition the density of the loop connecting H11 to H12 should also be shown.

Authors' response: We added a few new representations to **Supplementary Fig. 9** to hopefully better show that crystal packing does not influence the repressive helix 12 conformation in the orthosteric pocket—whereas the helix 12 conformations representative of most PPAR γ crystal structures show either helix 12/AF-2 interactions or a helix 12/helix 12 interaction (**Supplementary Fig. 8**). Admittedly, this is a difficult concept to show in a figure to a large degree because helix 12 is not solvent exposed as most other PPAR γ crystal structures that show clear crystal packing helix 12 contacts and would therefore not be expected to make crystal packing contacts with adjacent molecules in the crystal. As requested, we included density representations for the helix 11-12 loop in **Supplementary Figs. 1–3**.

4. To validate the functional importance of the repressive conformation, the authors mutated various AA and analyze their effect on repressor interactions. They should also show that the PPAR γ mutations prevent corepressor in the context of full-length proteins by mammalian 2-hybrid assay for example.

Authors' response: In this revision, we have included two new cellular functional assays. We assessed the effect of PPAR γ mutations within the context of full-length PPAR γ transcription using a cell-based reporter assay (**Figs. 5d and 6i**). We also assessed the impact of select mutations on the interaction between the PPAR γ LBD and entire NCoR RID using the mammalian two-hybrid assay (**Fig. 6h**). As described in the revised text,

| all of these data are consistent with the FP and TR-FRET biochemical assay data and structural studies using corepressor peptides.

Minor comments

-Page 2. Line 47: "...some of which some..."

| Authors' response: We fixed this error.

-Page 4. Line 101-109. The active conformation of PPAR γ is already known. This paragraph should be shorten.

| Authors' response: We acknowledge that the active conformation of PPAR γ is already known. We include a relatively short paragraph because we think this information is important for readers that may not be completely familiar with the active conformation to set the stage for describing the repressive conformation in the next paragraph in more detail.

-Page 9, line 256-257: This sentence is unclear.

| Authors' response: We extensively modified the discussion section and changed this sentence to read: "Our work here also describes the conformational ensemble of apo-PPAR γ helix 12 in solution at atomic resolution."

-Crystal structures validation: The validation reports suggest that following structures can be improved: 6ONI (Rfree, Ramachandran outliers), 6ONJ (side chain outliers, Ramachandran outliers), and 6PDZ (Ramachandran outliers)

| Authors' response: We performed additional refinement of the structures to improve quality. We included an updated crystallography table of statistics (**Supplementary Table 1**) and provide updated PDB validation reports with our revised submission.

REVIEWERS' COMMENTS:

Reviewer #1 (Remarks to the Author):

The authors have satisfactorily addressed all points raised by this reviewer at the previous review.

Reviewer #2 (Remarks to the Author):

What I meant by the comment 1 in the previous review is that it is better to use a couple of samples, each containing an MTSL at a different position, not a different spin label reagent. However, I understood that the authors had used PRE only to obtain qualitative data. The authors have addressed the other points raised by this reviewer.

Reviewer #3 (Remarks to the Author):

The authors have done a very thorough job in addressing the reviewers concerns and in providing the requested data.

Few precisions should be included for clarification:

-the crystal of PPAR γ -T0070907-SMRT contains 2 molecules in the asymmetric unit. Is the two complex identical? What about H12 conformation in the 2 molecules?

-the volume of the LBP doubles when bound to T0070907 and CoR. However, the figure seems to show that the pocket is partially open. It should be clarified.

Reviewer Comments

Reviewer #1

The authors have satisfactorily addressed all points raised by this reviewer at the previous review.

Reviewer #2

What I meant by the comment 1 in the previous review is that it is better to use a couple of samples, each containing an MTSL at a different position, not a different spin label reagent. However, I understood that the authors had used PRE only to obtain qualitative data. The authors have addressed the other points raised by this reviewer.

Authors' response: Thank you for clarifying. We believe that our placement of the MTSL label at position K474 on helix 12 is the optimal placement to assess the structural location of helix 12. In both the active and repressive conformations, K474 is solvent exposed and therefore the attached MTSL should not significantly perturb the structure. Additionally, when studying the apo-receptor, NMR peaks corresponding to helix 12 are not observed due to intermediate exchange on the NMR time scale. However, placement of MTSL on helix 12 via K474 enabled PRE measurements to determine the locations of helix 12 in regions where NMR peaks are observed—this cannot be done by placing the MTSL label at another location because PRE effects to helix 12 would not be observed due to the intermediate exchange. We thank the reviewer for understanding that our PRE data are only qualitative to sufficiently support our crystal structures.

Reviewer #3

The authors have done a very thorough job in addressing the reviewers' concerns and in providing the requested data.

Few precisions should be included for clarification:

-the crystal of PPAR γ -T0070907-SMRT contains 2 molecules in the asymmetric unit. Are the two complexes identical? What about H12 conformation in the 2 molecules?

Authors' response: There are slight structural changes in the two molecules in the asymmetric unit. We added a structural overlay of these changes in **Supplementary Fig. 2**.

-the volume of the LBP doubles when bound to T0070907 and CoR. However, the figure seems to show that the pocket is partially open. It should be clarified.

Authors' response: The repressive structures bound to T0070907 and NCoR do not have the Ω -loop modeled in due to poor density. It is possible that this gives the appearance that the pocket is partially open in our figures. This is why we included a pocket volume calculation with and without the Ω -loop for the active conformation crystal structure, which has the Ω -loop modeled in.